# Less atmospheric radiative heating by dust due to the synergy of coarser size and aspherical shape

Akinori Ito[1], Adeyemi A. Adebiyi[2,3], Yue Huang[2], and Jasper F. Kok[2]

[1]Yokohama Institute for Earth Sciences, JAMSTEC, Yokohama, Kanagawa, 236-0001, Japan.
[2]Department of Atmospheric and Oceanic Sciences, University of California, Los Angeles, CA 90095, USA.
[3]Department of Life and Environmental Sciences, University of California - Merced

*Correspondence to*: Akinori Ito (akinorii@jamstec.go.jp)

**Abstract.** Mineral dust aerosols cool and warm the atmosphere by scattering and absorbing solar (short-wave: SW) and thermal (long-wave: LW) radiation. However, significant uncertainties remain in dust radiative effects, largely due to differences in the dust size distribution and spectral optical properties simulated in Earth system models. Dust models typically underestimate the coarse dust load (more than 2.5 μm in a diameter) and assume a spherical shape, which leads to an overestimate of the fine dust load (less than 2.5 μm) after the dust emissions in the models are scaled to match observed dust aerosol optical depth at 550 nm ($DAOD_{550}$). Here, we improve the simulated dust properties with datasets that leverage measurements of size-resolved dust concentration, asphericity factor, and refractive index in a coupled global chemical transport model with a radiative transfer module. After the adjustment of size-resolved dust concentration and spectral optical properties, the global and annual average of $DAOD_{550}$ from the simulation increases from 0.023 to 0.029 and falls within the range of a semi-observationally-based estimate (0.030 ± 0.005). The reduction of fine dust load after the adjustment leads to a reduction of the SW cooling at the Top Of the Atmosphere (TOA). To improve agreement against a semi-observationally-based estimate of the radiative effect efficiency at TOA, we find that a less absorptive SW dust refractive index is required for coarser aspherical dust. Thus, only a minor difference is estimated for the net global dust radiative effect at TOA (–0.08 vs. –0.00 W·m$^{-2}$ on a global scale). Conversely, our sensitivity simulations reveal that the surface warming is substantially enhanced near the strong dust source regions (less cooling to –0.23 from –0.60 W·m$^{-2}$ on a global scale). Thus, less atmospheric radiative heating is estimated near the major source regions (less heating to 0.15 from 0.59 W·m$^{-2}$ on a global scale), because of enhanced LW warming at the surface by the synergy of coarser size and aspherical shape.

## 1 Introduction

Mineral dust aerosols can both cool and warm the climate, but how much dust aerosols net influence global climate is highly uncertain (Penner, 2019). Global dust modeling studies have suggested that mineral dust exerts global and annual mean aerosol radiative effect (RE) between –0.6 and +0.2 W m$^{-2}$ at the Top Of the Atmosphere (TOA) and between −0.2 and −2.7 W m$^{-2}$ at the surface (Miller and Tegen, 1998; Balkanski et al., 2007; Tanaka et al., 2007; Takemura et al., 2009; Räisänen et al., 2013; Zhao et al., 2013; Albani et al., 2014; Colarco et al., 2014; Heald et al., 2014; Di Biagio et al., 2020; Tuccella et al., 2020). Whereas a negative RE corresponds to the cooling of the global system when the sunlight is reflected to space, a positive RE corresponds to an overall warming of the Earth-atmosphere system by trapping incident short-wave (SW) and outgoing long-wave (LW) radiation. Radiative effect by dust aerosols perturbs surface temperature, wind speed, rainfall, and vegetation cover, which may induce

feedback on dust emissions (Perlwitz et al. 2001; Miller et al., 2004a; Colarco et al., 2014). The climate feedback does
not only depend on RE at TOA or the surface alone but also on the difference to the value at TOA and surface, which
represents radiative heating within the atmosphere (Miller et al., 2004b; Yoshioka et al., 2007; Lau et al., 2009). The
large uncertainties in quantifying the dust RE in the models are mainly propagated from the large spatial heterogeneity
and temporal variability of mineral dust abundance and the physicochemical properties (e.g., size distribution, mineral
composition, and shape), as well as the ground surface characteristics and atmospheric properties (e.g., surface
reflectance, temperature, and atmospheric absorption) (Sicard et al., 2014; Lacagnina et al., 2015; Li and Sokolik,
2018). The model errors in dust size distribution and particle shape can lead to an overestimate of fine dust load after
the dust emissions in the models are scaled to match observed dust aerosol optical depth at 550 nm ($DAOD_{550}$). The
corresponding overestimate of SW cooling might be compensated for in models by using a refractive index that is too
absorbing (Di Biagio et al., 2019, 2020), which depends on the mineral composition of the dust. We regard "fine" and
"coarse" dust as dust particles with a diameter less than 2.5 μm (i.e., $PM_{2.5}$) and between 2.5 and 20 μm, respectively.
Below, we provide a brief discussion of the effects of the dust size distribution, particle shape, and mineral
composition on dust radiative effects.

First, there has been increased attention paid to the importance of accurately predicting the abundance of coarse

dust for the global energy balance (Kok et al., 2017; Song et al., 2018; Di Biagio et al., 2020; Adebiyi and Kok, 2020).
The coarser particles are expected to be more prevalent closer to the source regions, as they fall much faster than finer
particles (Mahowald et al., 2014). For instance, the lifetime of dust aerosols larger than 30 μm in diameter is less than
12 h in most cases except in large haboobs (Ryder et al., 2013). Current models, however, cannot accurately simulate
observed transport of coarse dust particles across the Atlantic (Weinzierl et al., 2017; Ansmann et al., 2017), although
several hypotheses have been proposed to explain measurements of giant dust particles (larger than 63 μm in diameter)
relatively far from source regions (van der Does et al., 2018). The potential mechanism for long-range transport of
giant dust particles is that the uplift events of coarse dust can be induced by a nocturnal low-level jet or cold pool
outflow from mesoscale convective systems (i.e., haboobs) (Rosenberg et al., 2014; Ryder et al., 2019). At higher
elevation, electrostatic forces might retard the settling of coarse and giant dust particles and thus may facilitate the
transport of these particles over longer distances (Harrison et al., 2018; Toth et al., 2019). Other missing processes
that affect the transport and deposition of giant particles would also need to be incorporated into the models to
reproduce the measurements of the size distribution over the open ocean (van der Does et al., 2018). The coarse dust
particles scatter and absorb both the solar and thermal radiation, causing a net warming effect at TOA. In contrast, the
fine dust particles principally scatter SW radiation, causing a net cooling effect. Since coarse dust tends to warm the
climate, the underestimation of the abundance of coarse dust causes Earth system models to underestimate the
warming near the dust source regions.
Second, previous studies have shown that the SW radiative effect of dust asphericity on climate simulations is
minor on a global scale, partly because the larger DAOD is compensated for by the larger asymmetry parameter of
aspherical dust, which reduces the amount of radiation scattered backward to space (Räisänen et al., 2013; Colarco et
al., 2014). Moreover, non-spherical calcium-rich dust particles can be converted to spherical particles, due to
heterogeneous reactions with nitrate and sulfate on these particles, especially over polluted regions (Laskin et al.,
2005; Matsuki et al., 2005). As the plumes move downwind to the oceans, the dust aerosols can be aggregated with
sea salt in the marine boundary layer, which leads to more spherical shapes and larger sizes (Zhang and Iwasaka,
2004). However, the assumption of spherical shape in models leads to a substantial underestimation of the extinction
efficiency and thus DAOD near the strong source regions, mainly because the assumption of sphericity causes an
underestimation of the surface-to-volume ratio compared to aspherical dust (Kok et al., 2017, 2021; Hoshyaripour et
al., 2019; Tuccella et al., 2020). Radiative effect efficiency is often used for the evaluation of the models and is defined
as the gradient of a linear least squares fit applied to AOD and dust radiative effect at each two-dimensional (2-D)
grid box (W·m$^{-2}$ AOD$^{-1}$). Thus, the estimates of the dust radiative effect efficiency could be biased, in part, due to
large uncertainties associated with the spherical assumption on AOD retrieval (Zhou et al., 2020).
Third, the dust refractive index is often derived from measurements based on dust or individual mineral particles
(Bedidi and Cervelle, 1993; Long et al., 1993; Di Biagio et al., 2017, 2019; Stegmann & Yang, 2017). Indeed, most
dust particles are internal mixtures of various mineral compositions and irregular shapes (Reid, 2003; Wiegner et al.,
2009; Wagner et al., 2012). In desert soils, iron (Fe) oxides are generally hematite ($\alpha$-Fe$_2$O$_3$) and goethite (FeOOH),
which cause soil-derived dust absorption at ultraviolet (UV) and visible wavelengths (Sokolik and Toon, 1999;
Balkanski et al., 2007). These two minerals have distinct optical properties, which might cause various intensities of
SW absorption and thus RE of dust aerosols (Lafon et al., 2016). The dust complex refractive index in the LW also
depends on the particle mineralogical composition (Sokolik et al., 1998). Di Biagio et al. (2017) found a linear
relationship between the magnitude of the imaginary refractive index at 7.0, 9.2, and 11.4 μm and the mass
concentration of calcite and quartz absorbing at these wavelengths. However, the speciation of dust into its mineral
components inherently comprises uncertainties on soil mineralogy, mineral content in size-segregated dust particles,
and refractive index of mineral, partly due to the differences in prescribed parameters such as the particle size. The
atmospheric aging of Fe-containing aerosols can further modulate the optical properties of Fe oxides (Ito et al., 2018)
and organic carbon (Al-Abadleh, 2021), while the photochemical transformation of Fe oxides from lithogenic sources
due to atmospheric processing is relatively limited (< 10%), compared to pyrogenic sources (Ito et al., 2019).

| 97 | Here, we focus on the influence of the size-resolved abundance of aspherical dust on the aerosol radiative effects |
| 98 | in a coupled global chemical transport model (IMPACT) (Ito et al., 2020 and references therein) with a radiative |
| 99 | transfer module (RRTMG) (Iacono et al., 2008). We improve the accuracy of these simulations by correcting the bias |
| 100 | in size-resolved dust concentration with the Dust Constraints from joint Observational-Modelling-Experimental |
| 101 | analysis (DustCOMM) data set (Adebiyi et al., 2020), as well as by considering the aspherical shape (Huang et al., |
| 102 | 2020, 2021). We then explore the sensitivity to dust refractive index. |

**2 Methods**

We examined the dust radiative effects using ten combinations of different numerical experiments that varied (1) the simulated dust concentration and their size distribution, (2) particle shape, and (3) mineralogical composition (Tables 1 and 2). Two RRTMG calculations used the hourly averaged aerosol concentrations calculated from one IMPACT model simulation (E1 and E3) (denoted as "IMPACT"). The two sensitivity experiments were handled in the RRTMG calculations performed with the distinction between spherical and non-spherical dust and different refractive indices. We denoted "Sphere" when the RRTMG calculations used the spherical assumption on the particle shape, while the IMPACT model considered asphericity in calculation of gravitational settling velocities. On the other hand, we denoted "Asphere" when the dust asphericity was also considered in the RRTMG calculations. Subsequently, the simulated dust concentration and the size distribution were adjusted to the semi-observationally-based concentrations (Adebiyi and Kok, 2020) in another chemical transport model simulation, which was performed with the five RRTMG calculations (E4, E5, E6, E8, and E9) (denoted as "DustCOMM"). The term "semi-observationally-based" is used for DustCOMM, $DAOD_{550}$, and dust radiative effect efficiency when the estimates are based on the combination of observations and models. We examined different refractive indices for the dust mineralogy to represent the regional variations in refractive indices (denoted as "Mineral", "DB17", "DB19", "V83", "Less SW", "More LW", "More SW", and "Less LW"). Thus, the other three experiments (E2, E7, and E10) were calculated from the model output with a post-processor. DustCOMM-Asphere-DB19-V83 (E2) was obtained from combination of DustCOMM-Asphere-DB19-DB17 (E4) for SW and DustCOMM-Asphere-Mineral-V83 (E6) for LW. DustCOMM-Asphere-Less-More (E7) was obtained from combination of DustCOMM-Asphere-Less-Less (E8) for SW and DustCOMM-Asphere-More-More (E9) for LW. DustCOMM-Asphere-More-Less (E10) was obtained from combination of DustCOMM-Asphere-More-More (E9) for SW and DustCOMM-Asphere-Less-Less (E8) for LW. These sensitivity simulations and their radiative effects are summarized in Tables 1 and 2, respectively, with more details below. In section 2.3, we describe the DustCOMM data set used to adjust (1) size-resolved abundance of dust concentration. In section 2.4, we describe the adjustment factor of (2) particle shape for spectral optical properties. In section 2.5, we

describe differences in spectral refractive indices due to (3) different mineralogical compositions for the radiative flux
calculation.
**2.1      Aerosol chemistry transport model**

This study used the Integrated Massively Parallel Atmospheric Chemical Transport (IMPACT) model (Ito et

al., 2020 and references therein). Simulations were performed for the year 2016, using a horizontal resolution of
$2.0° \times 2.5°$ for latitude by longitude and 47 vertical layers. The chemical transport model was driven by the Modern
Era Retrospective analysis for Research and Applications 2 (MERRA-2) reanalysis meteorological data from the
National Aeronautics and Space Administration (NASA) Global Modeling and Assimilation Office (GMAO) (Gelaro
et al., 2017). Thus, the radiative feedback of the dust aerosol on the climate was not considered in this study.

The model simulated the emissions, chemistry, transport, radiation, and deposition of major aerosol species,

including mineral dust, black carbon (BC), particulate organic matter (POM), sulfate, nitrate, ammonium, and sea
spray aerosols, and their precursor gases. Dust emissions were dynamically simulated using a physically-based
emission scheme (Kok et al., 2014; Ito and Kok, 2017) with the soil mineralogical map (Journet et al., 2014; Ito and
Shi, 2016). Atmospheric processing of mineral dust aerosols, during transport, were projected for four distinct aerosol
size bins (<1.26 μm, 1.26–2.5 μm, 2.5–5 μm, and 5–20 μm of diameter). In this version of the IMPACT model, two
modes were used for sulfate aerosol (nuclei and accumulation mode), and two moments were predicted within each
mode (sulfate aerosol number and mass concentration) (Liu et al., 2005). The surface coating of sulfate on dust
aerosols occurred because of the condensation of sulfuric acid gas on their surfaces, coagulation with sulfate aerosol,
and formation in aqueous reactions within cloudy regions of the atmosphere (Liu et al., 2005). The heterogeneous
uptake of nitrate, ammonium, and water vapor by each aerosol for each size bin was interactively simulated in the
model following a hybrid dynamical approach (Feng and Penner, 2007). Five types of aerosols (i.e., dust, nucleated
sulfate, carbonaceous aerosols from fossil fuel combustion, carbonaceous aerosols from biomass burning, and sea salt)
were assumed to be externally mixed in each size bin for the computation of spectral optical properties (Xu and Penner,
2012). To derive atmospheric concentration of mineral composition for dust aerosol, "tagged" tracer was used for
each size-resolved mineral source. The direct emissions of dust were evenly distributed in mixing ratio throughout the
planetary boundary layer. The global scaling factor of dust emission was determined from the comparison of the model
results with ground-based AOD measurements near the dust source regions prior to the adjustment to the DustCOMM
(Kok et al., 2014; Ito and Kok, 2017). In recent review papers, multi-model evaluations of aerosol iron concentrations
and their solubilities have been comprehensively summarized on global and regional scales (Myriokefalitakis et al.,
2018; Ito et al., 2021).
To improve the accuracy of our simulations of mineral dust, we made several upgrades to the on-line emission
and gravitational settling schemes used in Ito et al. (2020). The dust emissions were extremely sensitive to soil
moisture, and thus the bias was adjusted with satellite observations (Ito and Kok, 2017). However, the satellite
measurements were only available every other day, depending on location. The Soil Moisture Active Passive (SMAP)
Level-4 Soil Moisture data product addressed these limitations by merging the satellite observations into a numerical
model of the land surface water and energy balance while considering the uncertainty of the observations and model
estimates (Reichle et al., 2019). In this work, we utilized the 3-hourly data of soil moisture derived from the SMAP
for barren and open shrublands separately (Reichle et al., 2018). To achieve this, we used the MODerate resolution
Imaging Spectroradiometer (MODIS) land cover map at 500 m resolution to calculate the fraction of barren and open
shrublands in each grand surface layer (Friedl et al., 2019)
Compared to the assumption on spherical shapes of aerosols, the dust asphericity increased aerodynamic drag
at a given volume and mass, and thus increased gravitational settling lifetime by about 20% (Huang et al., 2020). Here,
we implemented a globally averaged asphericity factor of 0.87 (Huang et al., 2020) to the gravitational settling scheme
for mineral dust. Nevertheless, the lifetime of the dust aerosol for the largest-size bin in the IMPACT model, even
after accounting for asphericity (1.4 days for 5–20 μm of diameter), was significantly shorter than an ensemble of
model results (2.1 ± 0.3 days for the mass mean diameter of 8.3 μm) (Kok et al., 2017). The impact of this
underestimate of atmospheric lifetime is explored using the DustCOMM data set, as was summarized in Table 2 (E3
– E4).
**2.2     Integration of IMPACT and RRTMG**
To improve the accuracy of our simulations of dust RE, we made upgrades to the radiative transfer calculations
(Ito et al., 2018 and references therein). In this study, we integrated the Rapid Radiative Transfer Model for GCMs
(RRTMG) online within the IMPACT model to calculate the radiative fluxes associated with atmospheric aerosols.
RRTMG is a radiative transfer code that calculates the SW and LW atmospheric fluxes (Iacono et al., 2008). Given
the size range of dust particles, scattering and absorption in the on-line model were described in terms of Mie theory.
Assuming homogeneous spherical particles, the spectral optical properties such as the mass extinction coefficient,
single scattering albedo, and asymmetry parameter were calculated using a look-up table as a function of refractive
index and size parameter (Xu and Penner, 2012). The impact of this spherical assumption is explored using aspherical
factor, as was summarized in Table 2 (E5 – E4).
The mineral dust particles were assumed to follow prescribed size distributions within each size bin (Liu et al.,
2005). In applying the look-up table, the size spectrum for mineral dust was divided into 30 sub bins (Wang and

Penner, 2009). As for the SW, the particle size increased with the uptake of sulfate, nitrate, ammonium, and water by the aerosols (Xu and Penner, 2012). These coating materials on aerosol cores were treated as internally mixed with each aerosol core in each size bin. Thus, the coating materials on dust only can reduce solar absorption of mineral dust. Subsequently, these optical properties were used by the RRTMG to calculate RE based on dust mixing ratio distributions in the IMPACT model. The dust RE was estimated as the difference in the calculated radiative fluxes with all aerosols and with all aerosols except the dust aerosols coated with sulfate, nitrate, ammonium, and water for each bin. As the LW scattering was not accounted for in the RRTMG, we multiplied the LW radiative fluxes by the adjustment factors of $1.18 \pm 0.01$ and $2.04 \pm 0.18$ for the dry particles at the surface and TOA (Dufresne et al., 2002), following Di Biagio et al. (2020). The larger adjustment factor at TOA reflects the fact that the upward LW radiation emitted from the ground surface can be trapped through scattering and absorption compared to the surface.

The broadband direct and diffuse albedos for both the UV visible and visible IR were specified from the hourly MERRA-2. The surface emissivity was based on the hourly MERRA-2. Long-lived greenhouse gas concentrations were obtained from historical greenhouse gas concentrations for climate models (Meinshausen et al., 2017). Water vapor concentrations were specified according to the MERRA-2. Cloud optical properties were calculated based on the liquid and ice visible optical depths from the MERRA-2, prescribing effective radii of 10 μm for water droplets and 25 μm for ice particles, respectively (Gettelman et al., 2010; Heald et al., 2014).

**2.3    DustCOMM dataset and sensitivity experiments to size-resolved dust concentration**

Dust Constraints from joint Observational-Modelling-experiMental analysis (DustCOMM) is a dataset of three-dimensional (3-D) dust properties obtained by combining observational, experimental, and modeling constraints on dust properties. While details can be found in Adebiyi et al. (2020) and Adebiyi and Kok (2020), we provide a brief overview here. First, DustCOMM's constraint on the 3-D dust size distribution combines dozens of previously published in-situ measurements of dust size distributions taken during several field campaigns, with an ensemble of climate model simulations. The framework used those in-situ measurements first to constrain the globally averaged size distribution (Adebiyi and Kok, 2020), which is used subsequently to adjust the bias in an ensemble of six global model simulations (Adebiyi et al., 2020). The constraints on dust size distribution range from 0.2 μm to 20 μm in diameter, where a generalized analytical function describes the sub-bin distribution based on brittle fragmentation theory (Kok, 2011). The second DustCOMM product – atmospheric dust mass loading – combines the constraints on dust size distribution with constraints on dust extinction efficiency and dust aerosol optical depth (Adebiyi et al., 2020). The constraints on dust extinction efficiency used the single-scattering database of Meng et al. (2010) and leveraged measurements of the dust index of refraction as well as accounts for the non-spherical shape of dust particles (Kok et

al., 2017). For this, we approximate dust as tri-axial ellipsoidal particles described by the globally representative
values of measured dust aspect ratio (the length-to-width ratio), and the height-to-width ratio (HWR) obtained from
Huang et al. (2020). Furthermore, the dust aerosol optical depth used to obtain the dust mass loading combines the
semi-observationally-based dataset from Ridley et al. (2016) with information from four reanalysis products. This
includes the MERRA-2, Navy Aerosol Analysis and Prediction System (NAAPS), Japanese Reanalysis for Aerosol
(JRAero), and Copernicus Atmosphere Monitoring Service interim ReAnalysis (CAMSiRA) (Adebiyi et al., 2020).
The aerosol RE of mineral dust strongly depends on both the magnitude of dust load and the dust size distribution
(Tegen and Lacis, 1996; Liao and Seinfeld, 1998). The DustCOMM data set contains total column loading (X, Y) and
concentration of mineral dust resolved by season (T) and particle size (S) (Adebiyi et al., 2020). To correct the bias in
the seasonally averaged size-resolved dust emission in the IMPACT model, $E_{IMPACT}(X, Y, T, S)$, the sum of bin 1, bin
2, and bin 3 dust emission flux was scaled by the seasonal mean of the ratio of the sum of bin 1, bin 2, and bin 3 dust
column loading between the model, $L_{IMPACT}(X, Y, T, S)$, and DustCOMM, $L_{DustCOMM}(X, Y, T, S)$, at each 2-D grid
box. The bias correction factor, $L_{bias}(X, Y, T)$, between the IMPACT model and DustCOMM data set is given by:
$$L_{bias}(X, Y, T) = \sum_{S=1}^{3} L_{DustCOMM}(X, Y, T, S) \div \sum_{S=1}^{3} L_{IMPACT}(X, Y, T, S) \ (1).$$

When the source function was used for high-latitude dust in the Northern Hemisphere, this led to substantially high
emissions and thus RE over there, likely due to the influences from long-range transported dust. Therefore, the direct
emissions of dust from the nine major source regions only (Kok et al., 2021) were adjusted using the DustCOMM
data (Fig. 1). To adjust the size bias in dust emissions, the mass fraction of emitted dust for each bin was prescribed
according to the size-resolved total column loading of DustCOMM at each 2-D grid box. The mass fraction for each
size bin, $S_{DustCOMM}(X, Y, T, S)$ is given by:
$$S_{DustCOMM}(X, Y, T, S) = L_{DustCOMM}(X, Y, T, S) \div \sum_{S=1}^{4} L_{DustCOMM}(X, Y, T, S) \ (2).$$

Thus, the dust emission flux after the adjustment, $E_{DustCOMM}(X, Y, T, S)$ is given by:
$$E_{DustCOMM}(X, Y, T, S) = L_{bias}(X, Y, T) \times S_{DustCOMM}(X, Y, T, S) \times E_{IMPACT}(X, Y, T) \ (3).$$

Overall, the IMPACT-simulated lifetime of the dust aerosol for the second-size bin (7.8 days 1.26–2.5 µm of diameter)
was in good agreement with the ensemble of model results (8.5 ± 1.1 days for the mass mean diameter of 1.8 µm)
(Kok et al., 2017). To correct the bias in the seasonally averaged 3-D dust size distribution after the transport, the mass
fraction of dust concentration for each bin between 0.2 and 20.0 µm of diameter was scaled at each 3-D grid box prior
to calculating the radiative fluxes using the RRTMG by the ratio of mass concentration of $PM_{2.5}$ (i.e., the sum of bin
1 and bin 2) to each bin (Table 3).

**2.4    Asphericity factor for optical properties and sensitivity experiments to particle shape**

To account for the dust asphericity, an adjustment factor was applied to the spherical optical properties at each dust size parameter and refractive index. The adjustment factors for the spectral optical properties of non-spherical particles were calculated after Huang et al. (2021). The atmospheric aging of mineral dust can form a uniform coating around the mineral core and therefore decrease particle asphericity during transport. This is implicitly considered in the globally averaged shape distribution of dust (Huang et al., 2019). Specifically, Huang et al. (2021) combined globally representative dust shape distributions (Huang et al., 2020) with a shape-resolved single-scattering database (Meng et al., 2010). This database combines four computational methods (Mie theory, T-matrix method, discrete dipole approximation, and an improved geometric optics method) to compute the single-scattering properties of non-spherical dust for a wide range of shape descriptors. Huang et al. (2021) provided the look-up table containing optical properties of non-spherical dust as functions of size parameter and refractive index.

The approximation of particles to spheres is evaluated by applying aspherical factors to the optical properties of the mass extinction coefficient, single scattering albedo, and asymmetry parameter for SW, as well as absorption fraction of extinction for the LW. At the same time, we maintained the consideration of asphericity on the gravitational velocity and kept the dust concentrations unaltered between the spherical (denoted as "Sphere") and aspherical (denoted as "Asphere") cases.

**2.5    Spectral refractive index and sensitivity experiments to mineralogical compositions**

The aerosol RE of mineral dust depends on mineralogical composition. For the sensitivity simulation to the SW and LW refractive indices, we used the global mean of laboratory measurements of the refractive index from 19 natural soils from various source regions around the world in Di Biagio et al. (2019) (denoted as "DB19") and in Di Biagio et al. (2017) (denoted as "DB17"), respectively. To illustrate the regional heterogeneity of refractive index, the refractive index obtained from 19 samples was aggregated into 9 main source regions, and the arithmetic mean was calculated for each source region (Di Biagio et al., 2017, 2019). The regionally averaged imaginary parts of the refractive indices at the wavelength of 0.52 μm and 9.7 μm showed large differences in SW and LW absorptivity, respectively, between different samples collected at various geographical locations (Fig. 1).

The optical properties from the measurements for dust samples generated from 19 natural soils suggested a considerable role of Fe oxides in determining the SW absorption (Di Biagio et al., 2019). The refractive indices for mineral components were used for hematite, goethite (Bedidi and Cervelle, 1993), silicate particle group, quarts, gypsum ($CaSO_4$) (Stegmann & Yang, 2017), and calcite ($CaCO_3$) (Long et al., 1993) in the simulations denoted as "Mineral". The hematite and goethite were treated separately according to the mineralogical map (Journet et al., 2014).

Consequently, hematite mass content averaged in the dust at emission (0.79% for fine and 0.50% for coarse from the
IMPACT simulation) was lower than goethite content (1.8% and 1.3%, respectively) on a global scale. In addition to
the primary emission of gypsum, $CaSO_4$ is secondarily formed due to the dissolution/precipitation of $CaCO_3$ in
thermodynamic equilibrium condition (Ito and Feng, 2010). To illustrate the difference in refractive index, the global
mean of the mineral composition was used for the comparison with DB19 (Fig. 1). The imaginary parts of the
refractive indices from mineralogical map were higher than DB19, resulting in a stronger absorption over the SW
spectrum.
The mineral dust LW refractive index also depends on its mineralogical composition (Sokolik et al., 1998; Di
Biagio et al., 2017). The LW refractive index of Volz (1983) has been widely used in climate models and satellite
remote sensing algorithms and thus was examined here (denoted as "V83") (Song et al., 2018). The imaginary parts
of the refractive indices from V83 were higher than DB17, resulting in a stronger absorption over most of the LW
spectrum. To analyze the dependence of the results on less (more) absorptive SW and less (more) absorptive LW
refractive indices, we made further sensitivity simulations by varying the values of imaginary parts of the refractive
index within the range of values from Di Biagio et al. (2017, 2019) (10% or 90% percentiles for SW or LW,
respectively) (denoted as "Less" or "More"). The associated real parts with 10% or 90% percentile imaginary parts
for LW were calculated to account for the Kramers-Kronig relation (Lucarini et al., 2005).
**2.6 Semi-observationally-based dust SW and LW radiative effect efficiency**
To estimate dust radiative effect efficiency, aerosol and radiation remote sensing products have been used with
various methods (Table 4) (Zhang and Christopher 2003; Li et al. 2004; Christopher and Jones 2007; Brindley and
Russell 2009; Yang et al. 2009; Di Biagio et al. 2010; Hansell et al. 2010; Hansell et al. 2012; Song et al. 2018).
The instantaneous SW radiative effect efficiency at TOA is obtained from the linear regression of TOA
radiation flux versus AOD observations, although the values in low-dust periods can be substantially influenced by
other types of aerosols such as biomass burning (Li et al. 2004). This radiative effect efficiency corresponds to the
instantaneous value derived under the limited condition at the measurements (e.g., solar position, atmospheric
condition). From the extrapolation of the instantaneous value, the diurnal mean dust SW radiative effect efficiency at
the surface and TOA can be derived based on model calculations.
The LW radiative effect efficiency at TOA can be obtained from the linear regression of TOA radiation flux
versus AOD observations over the source regions (Brindley and Russell 2009). However, the observed outgoing LW
radiation is not only dependent on DAOD but also on other factors such as dust layer height, water vapor content, and
other types of aerosols. Thus, the LW radiative effect efficiency is estimated from the difference between observed
outgoing LW radiation and the dust-free outgoing LW radiation, which can be estimated using radiative transfer model
(Song et al., 2018).
Consequently, the semi-observationally-based estimates of the dust radiative effect efficiency could be biased,
in part, due to large uncertainties associated with the estimation method, the selection of cloud-free and dust-dominant
data, and dust physicochemical properties. To understand the sensitivity of the dust radiative effect efficiency to the
particle size distribution, asphericity, and refractive index of dust, radiative transfer computations have been carried
out in previous studies (Li et al., 2004; Song et al., 2018). Song et al. (2018) found that the combination of the coarser
dust particle size distribution and the more absorptive LW refractive index (V83) yielded the best simulation of the
dust LW radiative effect in comparison with the satellite flux observations (i.e., Clouds and the Earth's Radiant Energy
System (CERES)), compared to the less absorptive LW refractive index (DB17).
**3.    Results and Discussions**
We evaluate our results from the sensitivity simulations against semi-observationally-based estimates of
$DAOD_{550}$ in section 3.1 and radiative effect efficiency for SW and LW in section 3.2 and section 3.3, respectively.
We focus this evaluation on the North Africa and the North Atlantic in boreal summer (June, July, and August) partly
because that is the region and season for which most observational constraints on dust radiative effects are available.
The better agreement is obtained for the less absorptive SW (Di Biagio et al., 2019) and the more absorptive LW
(Volz, 1983) dust refractive indices with adjustments of size-resolved dust concentration and particle shape. Our
improved simulation from IMPACT-Sphere-Mineral-V83 (E1) to DustCOMM-Asphere-DB19-V83 (E2)
substantially reduces the model estimates of atmospheric radiative heating by mineral dust near the major source
regions even though it induces only a minor difference in RE at TOA on a global scale (section 3.4). To elucidate the
differences in dust radiative effects between different simulations, the results from the sensitivity simulations in
conjunction with previous modeling studies are analyzed in section 3.5.
**3.1    Dust load and aerosol optical depth**
We compared our model estimates of $DAOD_{550}$ against semi-observationally-based data in box plots and
Taylor diagrams (Taylor, 2001) for the evaluation of the various model experiments against semi-observationally-
based estimates (Ridley et al., 2016; Adebiyi et al., 2020) to provide a concise statistical summary of the bias,
correlation coefficient, root mean square errors, and the ratio of standard deviation (Fig. 2, Tables S1 and S2).
IMPACT-Sphere-Mineral-V83 (E1) simulations resulted in a significant underestimation of the global and annual
mean of $DAOD_{550}$ (0.023) (Fig. 2 and Table 3). After considering the dust asphericity for spectral optical properties,
we adjusted IMPACT-simulated dust loads against the constraints on dust load from the DustCOMM data set. This
adjustment led the simulated total dust load to increase from 25 Tg (E1) to 32 Tg (E2), which addressed the issue of
coarse dust underestimation and fine dust overestimation by the model (Fig. 3, Table 3). Consequently, the global and
annual mean of $DAOD_{550}$ from DustCOMM-Asphere-DB19-V83 (E2) simulation (0.029) fell within the range in the
semi-observationally-based estimate (0.030 ± 0.005) (Ridley et al., 2016) (Table 3). We found that the agreement in
the median with the semi-observationally-based estimate (0.127) was improved from IMPACT-Sphere-Mineral-DB17
(0.049) to DustCOMM-Asphere-DB19-V83 (0.117) (solid line within the box mark in Fig. 2d). We also found higher
$DAOD_{550}$ from E2 than E1 over East Asia and Bodele/Sudan in winter (Fig. 2, Table S2). The better agreement
suggested that DustCOMM-Asphere-DB19-V83 (E2) simulation was reasonably constrained by the $DAOD_{550}$ (Ridley
et al., 2016; Adebiyi et al., 2020).
**3.2     Dust SW radiative effect efficiency**
Modeled estimates of clear-sky dust SW radiative effect efficiencies ($W \cdot m^{-2} \, DAOD_{550}^{-1}$) at the surface (Table
S3) and TOA (Table S4) were compared with estimates reported by regional studies based on satellite observations
over the North Africa and the North Atlantic (Fig. 4). Sensitivity simulations demonstrated that the radiative effect
efficiency strongly depended on the particle size, refractive index, and particle shape (Fig. 4). The adjustment of size-
resolved dust concentration and shape with the same refractive index led to overestimates of the SW radiative effect
efficiencies against semi-observationally-based data at TOA (from E1 to E6 in Fig. 4h), because coarser dust absorbs
more SW radiation efficiently than finer particles. Subsequently, the use of less absorptive SW refractive index with
DustCOMM-Asphere-DB19-V83 (E2) simulations led to a better agreement (from E6 to E2 in Fig. 4). On the other
hand, the use of much less (10% percentile) absorptive SW refractive index from DustCOMM-Asphere-Less-More
(E7) simulation deteriorated the agreement due to the underestimate of cooling at the surface (Fig. 4g). In contrast,
the use of a more absorptive SW refractive index from DustCOMM-Asphere-Mineral-V83 (E6) improved the
agreement at the surface. However, the semi-observationally-based estimates of diurnally averaged radiative effect
efficiency at the surface were derived from extrapolation of the instantaneous values, which would affect the
comparison due to differences in the methodologies between dust models (section 2.6). The differences in the model-
based estimates of radiative effect efficiency might arise from different data sets of the refractive index, size
distribution, and particle shape (Song et al., 2018).
**3.3     Dust LW radiative effect efficiency**
Modeled estimates of clear-sky dust LW (Fig. 5) radiative effect efficiencies ($W \cdot m^{-2} \, DAOD_{550}^{-1}$) at the surface
(Table S5) and TOA (Table S6) were compared with estimates reported by regional studies based on satellite

observations over North Africa and the North Atlantic. Sensitivity simulations demonstrated that the radiative effect efficiency strongly depended on the particle size, refractive index, and particle shape (Fig. 5). Both the IMPACT-Sphere-Mineral-V83 (E1) and DustCOMM-Asphere-DB19-V83 (E2) simulations yielded better agreement with semi-observationally-based data at the surface and TOA, compared to the less absorptive LW dust refractive indices (E3, E4, E5, and E7) (Fig. 5). The relatively high LW radiative effect efficiencies over western Africa were also consistent with the semi-observationally-based data. On the other hand, the relatively low LW radiative effect efficiencies were found over eastern Africa. Moving toward the northeastern side of the region, however, the associated uncertainties in the semi-observationally-based values increased (Brindley and Russell 2009). The dust LW radiative effect efficiency depends strongly on the vertical profile of dust concentration, temperature, and water vapor, which would affect the comparison due to a high variability in these factors (section 2.6).

### 3.4 Less atmospheric radiative heating by dust due to the synergy of coarser size and aspherical shape

The Saharan dust cools the ground surface by reducing the solar radiation reaching the surface and warms the atmosphere by absorbing solar radiation (Fig. 6). On the other hand, thermal emission by dust warms the surface and cools the atmosphere (Fig. 7). Our sensitivity simulations showed that the annually averaged net instantaneous radiative effect due to mineral aerosol (NET) ranged from –0.48 (DustCOMM-Asphere-Less-Less) to +0.25 (DustCOMM-Asphere-Mineral-V83) $W \cdot m^{-2}$ at TOA (Table 5). The net RE from both the IMPACT-Sphere-Mineral-V83 ($-0.00$ $W \cdot m^{-2}$) and DustCOMM-Asphere-DB19-V83 ($-0.08$ $W \cdot m^{-2}$) simulations resulted within 98% confidential interval of DustCOMM data set ($-0.27$ to $0.14$ $W \cdot m^{-2}$).

The SW RE by dust outweighs the LW warming effect at the surface in the IMPACT-Sphere-Mineral-V83 (E1) simulation (Fig. 8). Consequently, the highly absorbing dust could play an important role in the aerosol radiative forcing for the climate models to alter the West African monsoon, with the radiative heating concentrated in the dust layer (Miller et al., 2004b; Lau et al., 2009). Our model results of dust RE from DustCOMM-Asphere-DB19-V83 (E2) simulation, however, suggested that the surface warming was substantially enhanced near the strong dust source regions ($-0.23$ $W \cdot m^{-2}$ on a global scale) (Fig. 8), compared to the IMPACT-Sphere-Mineral-V83 simulation ($-0.60$ $W \cdot m^{-2}$ on a global scale). Thus, our results demonstrated that the atmospheric radiative heating by mineral dust was substantially reduced for DustCOMM-Asphere-DB19-V83 (E2) simulation ($0.15$ $W \cdot m^{-2}$), compared to the IMPACT-Sphere-Mineral-V83 (E1) simulation ($0.59$ $W \cdot m^{-2}$).

## 3.5 Variability of dust radiative effect in different simulations

To elucidate the differences in dust radiative effects between the IMPACT-Sphere-Mineral-V83 (E1) and DustCOMM-Asphere-DB19-V83 (E2) simulations and to explore the variability in different previous model estimates (Fig. 9), the differences in annually averaged radiative effects of mineral dust from DustCOMM-Asphere-DB19-DB17 (E4) simulation were shown in Fig. 10. A slope of one in Fig. 10 represented an identical change in both the surface and TOA and thus corresponded to no change in radiative heating within the atmosphere. The distances from the DustCOMM-Asphere-DB19-DB17 (E4) simulation demonstrated that large uncertainties existed for the size distribution and spectral optical properties. Our sensitivity simulations revealed that the DustCOMM-Asphere-DB19-V83 (E2) simulation led to a similar net RE at TOA to the IMPACT-Sphere-Mineral-V83 (E1) simulation but resulted in less cooling at the surface (Fig. 9). This revision can be divided into (1) the size-resolved abundance (black hexagons, E3 – E4, in Fig. 10), (2) SW refractive index (red diamonds, E6 – E4, in Fig. 10), and (3) particle shape (red circles, E5– E4, in Fig. 10). Additionally, we show the sensitivity of dust RE to LW refractive index (DB17), which was used by both Di Biagio et al. (2020) and Balkanski et al. (2021).

First, at TOA, the SW RE was more sensitive to the size-resolved abundance (–0.17 W·m$^{-2}$ at the vertical axis of black hexagon in Fig. 10a), compared to LW (0.00 W·m$^{-2}$ at the vertical axis of black hexagon in Fig. 10b). Second, this less SW cooling effect with coarser dust (E3 – E4) was partially compensated for by more SW cooling with the use of the less absorptive SW refractive index (E4: –0.32 W·m$^{-2}$) than E6 (0.02 W·m$^{-2}$). Thirdly, the sensitivity of SW RE to dust asphericity was rather minor (0.04 W·m$^{-2}$ at the vertical axis of red circle in Fig. 10a), partly because the lower DAOD was compensated for by the lower asymmetry parameter of spherical dust, which enhanced the amount of radiation scattered backward to space (Räisänen et al., 2013; Colarco et al., 2014). The partial compensation led to a small enhancement of SW RE for the IMPACT-Sphere-Mineral-V83 (E1) simulation and thus the resulting similar net RE to DustCOMM-Asphere-DB19-V83 (E2) at TOA (Fig. 9).

In contrast, at the surface, our sensitivity simulations demonstrated substantially different responses in the RE, mostly because of LW warming effects (Fig. 9). The enhanced LW warming by coarser dust (–0.08 W·m$^{-2}$ at the horizontal axis of black hexagon in Fig. 10b) was accompanied by the asphericity (–0.15 W·m$^{-2}$ at the horizontal axis of red circle in Fig. 10b), because the enhancement of the absorption fraction of extinction due to asphericity was larger at coarser size. The enhanced LW warming effects of each as well as the synergy was further amplified using the more absorptive LW dust refractive index (Volz, 1983) (at the horizontal axis of red diamond in Fig. 10b). As a result, our sensitivity simulations revealed that substantially less dust absorption at LW due to the underestimation of the coarse dust load and the assumption of the spherical shape (IMPACT-Sphere-Mineral-V83) contributed to the less surface warming, compared to DustCOMM-Asphere-DB19-V83 (Fig. 9).

A relatively good agreement of net RE by dust at TOA with both Di Biagio et al. (2020) (–0.06 W·m$^{-2}$) and
Balkanski et al. (2021) (–0.02 W·m$^{-2}$) could be obtained from both the IMPACT-Sphere-Mineral-V83 (E1: –0.00
W·m$^{-2}$) and DustCOMM-Asphere-DB19-V83 (E2: –0.08 W·m$^{-2}$) simulations (Fig. 9 and Table 5). On the other hand,
our modeled dust net RE at the surface from DustCOMM-Asphere-DB19-V83 (E2: –0.23 W·m$^{-2}$) indicated much
less cooling than Di Biagio et al. (2020) (–0.63 W·m$^{-2}$), Balkanski et al. (2021) (–1.01 W·m$^{-2}$), and IMPACT-Sphere-
Mineral-V83 (E1: –0.60 W·m$^{-2}$). The synergy of coarser size and aspherical dust could contribute to the less surface
cooling of the DustCOMM-Asphere-DB19-V83 (E2), because of enhanced LW warming. At the same time, both Di
Biagio et al. (2020) and Balkanski et al. (2021) used DB17 and considered dust with diameters more than 20 μm.
Thus, the more absorptive LW dust refractive index (V83, E6 for LW: 1.00 W·m$^{-2}$) than DB17 (E4 for LW: 0.58
W·m$^{-2}$) (E6 – E4 for LW: 0.42 W·m$^{-2}$ at the horizontal axis of red diamond in Fig. 10b) could also contribute to the
less surface cooling, which might be partially compensated for in our model by the omission of dust with diameters
more than 20 μm. Consequently, our estimate of atmospheric radiative heating by dust from DustCOMM-Asphere-
DB19-V83 (E2: 0.15 W·m$^{-2}$) was lower than Di Biagio et al. (2020) (0.63 W·m$^{-2}$), Balkanski et al. (2021) (0.98
W·m$^{-2}$), and IMPACT-Sphere-Mineral-V83 (E1: 0.59 W·m$^{-2}$). Additionally, the hot and dry climate over brighter
desert surface exaggerates differences in RE at the surface between the models (Miller et al., 2014). The low humidity
allows dust particles to absorb LW radiation with reduced competition from water vapor, while high temperatures
within the boundary layer increase downward thermal emission by dust (Liao and Seinfeld, 1998). The reduction of
fine dust load after the adjustment leads to underestimates of the SW cooling at TOA. To improve agreement against
semi-observationally-based estimate of the radiative effect efficiency at TOA, the less absorptive SW dust refractive
index is required for coarser aspherical dust. Thus, uncertainties in the size-resolved dust concentration, particle shape,
and refractive index contribute to the diversity in the simulated dust RE at the surface.
**4.        Conclusions**

Accurate estimates of the size-resolved dust abundance, their spectral optical properties, and their seasonality

in regional and vertical scales provide a step towards a more reliable projection of the climatic feedback of mineral
aerosols. The radiative effect efficiency depends on numerous variables in model simulations, including the spatial
distribution and temporal variation of size-resolved dust concentrations, the mass extinction coefficient, single
scattering albedo, and asymmetry parameter of dust. Since the models typically underestimate the coarse dust load
and overestimate the fine dust load, the sensitivity to the aerosol absorptivity might be considerably different from
previous studies. Thus, the model results should be re-evaluated against semi-observationally-based estimate of the
DAOD$_{550}$ and dust radiative effect efficiency.

We improved the accuracy of the simulations by adjusting the bias in size-resolved aspherical dust

concentration with the DustCOMM data set. Alternatively, dust mineralogy might contribute to the underestimation

of modeled aerosol absorption compared to satellite observations (Lacagnina et al., 2015). This enhanced aerosol

absorption was examined by specifying the mineralogy with varying amounts of light-absorbing Fe oxides for SW.

The better agreement with the semi-observationally-based data of dust radiative effect efficiency was obtained using

the less absorptive SW dust refractive indices after the adjustments of dust sizes and shapes.

The diversity of modeled dust net RE at the surface ($-1.64$ W·m$^{-2}$ to $-0.20$ W·m$^{-2}$) is much larger than at TOA

($-0.01$ W·m$^{-2}$ to $-0.60$ W·m$^{-2}$), partly because the refractive index is optimized to obtain reasonable agreement against

satellite observations of TOA radiation flux (e.g., CERES). The uncertainties in the size-resolved dust concentration,

particle shape, and refractive index contribute to the model diversity at the surface. DustCOMM-Asphere-DB19-V83

(E2) simulation resulted in less cooling at the surface by the synergy of coarser size and aspherical shape, compared

to IMPACT-Sphere-Mineral-V83 (E1) simulation ($-0.23$ vs. $-0.60$ W·m$^{-2}$ on a global scale). Consequently, the

atmospheric heating due to mineral dust was substantially reduced for the DustCOMM-Asphere-DB19-V83 (E2)

simulation (0.15 W·m$^{-2}$), compared to the intensified atmospheric heating from the IMPACT-Sphere-Mineral-V83

(E1) simulation (0.59 W·m$^{-2}$). The less intensified atmospheric heating due to mineral dust could substantially modify

the vertical temperature profile in Earth system models and thus has important implications for the projection  of dust

feedback near the major source regions in the past and future climate changes (Kok et al., 2018). More accurate

estimates of semi-observationally-based dust SW and LW radiative effect efficiencies over strong dust source regions

are needed to narrow the uncertainty in the RE.

Currently, the model did not include dust particles above 20 μm, but a substantial fraction of airborne dust near

source regions may be above this threshold (Ryder et al., 2019). Moreover, such large particles can be transported to

higher altitudes and longer distances than the model prediction. The higher the dust layer resides, the larger the dust

LW RE at TOA is estimated under the clear-sky conditions (Liao and Seinfeld, 1998). Marine sediment traps, which

are located underneath the main Saharan dust plume in the Atlantic Ocean, suggest that giant particles are dominated

by platy mica and rounded quartz particles (van der Does et al., 2016). Thus, mineral composition of the giant particles

could be different from the aerosol samples generated from soils in the laboratory by Di Biagio et al. (2017), which

may reflect less absorbing LW refractive index of DB17 than V83. Indeed, the dust sample was collected for V83

from rainwater after strong wind. On the other hand, the contribution of the LW scattering might be underestimated

in the models, as Di Biagio et al. (2020) noted that the adjustment factor was estimated for dust of diameter less than

10 μm and thus might be a lower approximation of the LW scattering by coarse dust. Therefore, a better understanding

of the effect of such large particles beyond 20 μm and mineralogical composition on radiation balance remains a topic

of active research, given their potential to amplify the warming of the climate system. In such an extreme case as the
"Godzilla" dust storm over the North Africa and the tropical Atlantic in June 2020 (Francis et al., 2020), the dust
loading could be larger than that examined for this study, and our estimates of the warming effects might be
conservative during such events. However, to keep the giant particles in the atmosphere, the modeled deposition fluxes
should be reduced from the current model. Therefore, models should improve their ability to capture the evolution of
the dust size distribution as the plumes move downwind of the source regions.
**Code availability.**
The source code of the RRTMG has been obtained from the website at https://github.com/AER-RC/RRTMG_LW and
https://github.com/AER-RC/RRTMG_SW. The source code of the Kramers-Kronig relations has been obtained from
the website at https://www.mathworks.com/matlabcentral/fileexchange/8135-tools-for-data-analysis-in-optics-
acoustics-signal-processing. The source code of the Taylor diagram has been obtained from the web site at
https://www.mathworks.com/matlabcentral/fileexchange/20559-taylor-diagram.
**Data availability.**
SMAP data have been obtained from the website at https://nsidc.org/data/smap/smap-data.html. MODIS land data
have been retrieved from the website at https://ladsweb.modaps.eosdis.nasa.gov/. MERRA-2 data have been provided
by the Global Modeling and Assimilation Office (GMAO) at NASA Goddard Space Flight Center
(https://disc.gsfc.nasa.gov/datasets/). The DustCOMM data are available at https://dustcomm.atmos.ucla.edu/. The
datasets supporting the conclusions of this article are included within the article and its supplement file.
**Supplement.**
The supplement related to this article is available online at:
**Author contributions.**
AI and JFK initiated the modeling collaboration with semi-observationally-based data sets. AI carried out the
modeling study. AAA, YH, JFK contributed semi-observationally-based data sets of DustCOMM and asphericity
factor. All authors read and approved the final manuscript.
**Competing interests.**
The authors declare that they have no competing interests.

**Acknowledgements.**

Numerical simulations were performed using the Hewlett Packard Enterprise (HPE) Apollo at the Japan Agency for

Marine-Earth Science and Technology (JAMSTEC).

**Financial support.**

Support for this research was provided to A.I. by JSPS KAKENHI Grant Number 20H04329 and 18H04143, and

Integrated Research Program for Advancing Climate Models (TOUGOU) Grant Number JPMXD0717935715 from

the Ministry of Education, Culture, Sports, Science and Technology (MEXT), Japan. This work was developed with

support from the University of California President's Postdoctoral Fellowship awarded to A.A.A., and from the

National Science Foundation (NSF) grants 1552519 and 1856389 awarded to J.F.K. Y.H. acknowledges support from

NASA grant 80NSSC19K1346, awarded under the Future Investigators in NASA Earth and Space Science and

Technology (FINESST) program.

Review statement.

This paper was edited by Susannah Burrows and reviewed by four anonymous referees.

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

 **Figure captions**

Figure 1.     Imaginary part of the refractive index at (a) 0.52 μm, (b) SW, (c) 9.7 μm, and (d) LW. The refractive
index obtained from 19 samples was aggregated into 9 main source regions and the arithmetic mean was
calculated for each source region (Di Biagio et al., 2017, 2019). The global mean is used for others. The
coordinates of the nine source regions were: (S1) western North Africa (20°W – 7.5°E; 18°N – 37.5°N), (S2)
eastern North Africa (7.5°E – 35°E; 18°N – 37.5°N), (S3) the Sahel (20°W – 35°E; 0°N – 18°N), (S4) Middle
East / Central Asia (30°E – 70°E for 0°N – 35°N, and 30°E – 75°E for 35°N – 50°N), (S5) East Asia (70°E
– 120°E; 35°N – 50°N), (S6) North America (130°W – 80°W; 20°N – 45°N), (S7) Australia (110°E – 160°E;
10°S – 40°S), (S8) South America (80°W – 20°W; 0°S – 60°S), and (S9) Southern Africa (0°E – 40°E; 0°S
– 40°S).
Figure 2.     The model better reproduced semi-observationally-based data of $DAOD_{550}$ after adjusting the size-
resolved dust load with DustCOMM and considering the dust asphericity. (a) semi-observationally-based
estimates of the $DAOD_{550}$ were averaged over 2004–2008 (Ridley et al., 2016; Adebiyi et al., 2020). The
annually averaged model results were shown for (b) DustCOMM-Asphere-DB19-V83 (E2) and (c) the
differences between IMPACT-Sphere-Mineral-V83 (E1) and E2 simulations. (d) Comparison of seasonally
averaged $DAOD_{550}$ for semi-observationally-based (SOB) data, E1, E2, IMPACT-Asphere-DB19-DB17
(E3), and DustCOMM-Sphere-DB19-DB17 (E5). The square symbol represents the mean. The solid line
within the box mark shows the median. The boundaries of the box mark the 25th and 75th percentiles. The
whiskers above and below the box indicate the 1.5 × interquartile range, and the points indicate the outside
of the range. (e) Taylor diagram summarizing the statistics of the comparison against the seasonally averaged
regional $DAOD_{550}$ for the different experiments. The horizontal axis shows the standard deviation of the data
set or model prediction, the curved axis shows the correlation, and the green dashed lines denote the root-
mean-squared errors between the semi-observationally-based data and the model predictions. As such, the
distance between the semi-observationally-based data and the model predictions is a measure of the model's
ability to reproduce the spatiotemporal variability in the semi-observationally-based data. The coordinates
and the values of $DAOD_{550}$ at the 15 regions (marked in Fig. 3a) in summer were listed in Table S1. The
comparison for other seasons was presented in Table S2.
Figure 3.     Model-simulated dust loads at fine (smaller than 2.5 μm of diameter) and coarse size ranges (larger
than 2.5 μm of diameter) before and after adjusting the size-resolved dust load with DustCOMM. Results
were shown for (a) fine dust from DustCOMM, (b) fine dust from IMPACT-Sphere-Mineral-V83 (E1), (c)
fine dust from DustCOMM-Asphere-DB19-V83 (E2), (d) coarse dust from DustCOMM, (e) coarse dust from

841   E1, and (f) coarse dust from E2 simulations. The parentheses represented the global dust burden (Tg). The

842   values of dust load at each bin were listed in Table 3.

843 Figure 4.   Dust clear-sky SW radiative effect efficiency (W·m$^{-2}$ DAOD$^{-1}$). Semi-observationally-based data

844   at (a) the surface and (b) TOA were based on satellite observations (Yang et al. 2009; Li et al., 2004: Song

845   et al., 2018; Christopher and Jones, 2007). The model results were shown for (c) and (d) IMPACT-Sphere-

846   Mineral-V83 (E1), and (e) and (f) DustCOMM-Asphere-DB19-V83 (E2) simulations at the surface and TOA,

847   respectively. Comparison of seasonally averaged SW radiative effect efficiency for semi-observationally-

848   based (SOB) data and the different experiments at (g) the surface and (h) TOA. The square symbol represents

849   the mean. The solid line within the box mark shows the median. The boundaries of the box mark the 25th

850   and 75th percentiles. The whiskers above and below the box indicate the 1.5 × interquartile range, and the

851   points indicate the outside of the range. Taylor diagram summarizing the statistics of the comparison against

852   the seasonally averaged regional SW radiative effect efficiency for the different experiments at (i) the surface

853   and (j) TOA. The horizontal axis shows the standard deviation of the data set or model prediction, the curved

854   axis shows the correlation, and the green dashed lines denote the root-mean-squared errors between the semi-

855   observationally-based data and the model predictions. As such, the distance between the semi-

856   observationally-based data and the model predictions is a measure of the model's ability to reproduce the

857   spatiotemporal variability in the semi-observationally-based data. The regionally averaged values were listed

858   in Tables S3 and S4 at the surface and TOA, respectively.

859 Figure 5.   Dust clear-sky LW radiative effect efficiency (W·m$^{-2}$ DAOD$^{-1}$). Semi-observationally-based

860   estimates at (a) surface and (b) TOA were based on satellite observations (Song et al., 2018; Christopher and

861   Jones, 2007; Zhang and Christopher, 2003; Brindley and Russell, 2009; Yang et al., 2009). The model results

862   were shown for (c) and (d) IMPACT-Sphere-Mineral-V83 (E1), and (e) and (f) DustCOMM-Asphere-DB19-

863   V83 (E2) simulations at the surface and TOA, respectively. Comparison of seasonally averaged LW radiative

864   effect efficiency for semi-observationally-based (SOB) data and the different experiments at (g) the surface

865   and (h) TOA. The square symbol represents the mean. The solid line within the box mark shows the median.

866   The boundaries of the box mark the 25th and 75th percentiles. The whiskers above and below the box indicate

867   the 1.5 × interquartile range, and the points indicate the outside of the range. Taylor diagram summarizing

868   the statistics of the comparison against the seasonally averaged regional SW radiative effect efficiency for

869   the different experiments at (i) the surface and (j) TOA. The horizontal axis shows the standard deviation of

870   the data set or model prediction, the curved axis shows the correlation, and the green dashed lines denote the

871   root-mean-squared errors between the semi-observationally-based data and the model predictions. As such,

the distance between the semi-observationally-based data and the model predictions is a measure of the
model's ability to reproduce the spatiotemporal variability in the semi-observationally-based data. The
regionally averaged values were listed in Tables S5 and S6 at the surface and TOA, respectively.
Figure 6.     Dust SW radiative effect ($W \cdot m^{-2}$) and radiative heating of the atmosphere (i.e., the subtraction of
radiative effects from TOA to the surface in unit of $W \cdot m^{-2}$). The model results were shown for the simulations
for (a) IMPACT-Sphere-Mineral-V83 (E1) at the surface, (b) DustCOMM-Asphere-DB19-V83 (E2) at the
surface, (c) E1 in atmospheric column, (d) E2 in atmospheric column, (e) E1 at TOA, and (f) E2 simulations
at TOA. The numbers in parentheses represented the global mean.
Figure 7.     Dust LW radiative effect ($W \cdot m^{-2}$) and radiative heating of the atmosphere (i.e., the subtraction of
radiative effects from TOA to the surface in unit of $W \cdot m^{-2}$). The model results were shown for the simulations
for (a) IMPACT-Sphere-Mineral-V83 (E1) at the surface, (b) DustCOMM-Asphere-DB19-V83 (E2) at the
surface, (c) E1 in atmospheric column, (d) E2 in atmospheric column, (e) E1 at TOA, and (f) E2 simulations
at TOA. The numbers in parentheses represented the global mean.
Figure 8.     Dust net radiative effect ($W \cdot m^{-2}$) and radiative heating of the atmosphere (i.e., the subtraction of
radiative effects from TOA to the surface in unit of $W \cdot m^{-2}$). The model results were shown for the simulations
for (a) IMPACT-Sphere-Mineral-V83 (E1) at the surface, (b) DustCOMM-Asphere-DB19-V83 (E2) at the
surface, (c) E1 in atmospheric column, (d) E2 in atmospheric column, (e) E1 at TOA, and (f) E2 simulations
at TOA. The numbers in parentheses represented the global mean.
Figure 9.     Variability of dust radiative effect ($W \cdot m^{-2}$) in different model simulations at the surface and TOA
for (a) total dust SW, (b) total dust LW, and (c) total dust NET. The annually averaged values were listed in
Table 5.
Figure 10.     Radiative effect ($W \cdot m^{-2}$) of mineral dust due to various aerosol absorptivity at the surface and TOA
for (a) total dust SW, (b) total dust LW, and (c) total dust NET. The annually averaged values were listed in
Table 5. The dashed line represented a 1 : 1 correspondence and corresponded to no change in radiative
heating within the atmosphere.

**Table 1.** Summary of ten combinations of different numerical experiments compared in this study.

| Number | Experiment | Size-resolved dust | Sphericity | SW refractive index | LW refractive index |
|---|---|---|---|---|---|
| E1 | IMPACT-Sphere-Mineral-V83 | IMPACT | Sphere | Mineralogical map[d] | Volz (1983) |
| E2[a] | DustCOMM-Asphere-DB19-V83 | DustCOMM[b] | Asphere[c] | Di Biagio et al. (2019) | Volz (1983) |
| E3 | IMPACT-Asphere-DB19-DB17 | IMPACT | Asphere[c] | Di Biagio et al. (2019) | Di Biagio et al. (2017) |
| E4 | DustCOMM-Asphere-DB19-DB17 | DustCOMM[b] | Asphere[c] | Di Biagio et al. (2019) | Di Biagio et al. (2017) |
| E5 | DustCOMM-Sphere-DB19-DB17 | DustCOMM[b] | Sphere | Di Biagio et al. (2019) | Di Biagio et al. (2017) |
| E6 | DustCOMM-Asphere-Mineral-V83 | DustCOMM[b] | Asphere[c] | Mineralogical map[d] | Volz (1983) |
| E7 | DustCOMM-Asphere-Less-More | DustCOMM[b] | Asphere[c] | Less SW[e] | More LW[g] |
| E8 | DustCOMM-Asphere-Less-Less | DustCOMM[b] | Asphere[c] | Less SW[e] | Less LW[h] |
| E9 | DustCOMM-Asphere-More-More | DustCOMM[b] | Asphere[c] | More SW[f] | More LW[g] |
| E10 | DustCOMM-Asphere-More-Less | DustCOMM[b] | Asphere[c] | More SW[f] | Less LW[h] |

[a]Combination of DustCOMM-Asphere-DB19-DB17 (E4) for SW and DustCOMM-Asphere-Mineral-V83 (E6) for LW.

[b]Size-resolved dust concentration was adjusted with semi-observationally-based estimate (Adebiyi & Kok, 2020).

[c]Dust asphericity was considered in calculating the optical properties, which further assumed internal mixing of minerals (Huang et al., 2021) using a volume-weighted mixture for each size bin.

[d]Mineralogical composition of dust aerosol for each size was prescribed at emission by mineralogical map (Journet et al., 2014; Ito and Shi 2016). The more absorptive SW refractive indices (Bedidi and Cervelle, 1993; Stegmann & Yang, 2017; Long et al., 1993) were used for mineral dust, compared to the less absorptive global mean data set (Di Biagio et al., 2019).

[e]Less absorptive SW refractive indies were calculated by varying the values of the imaginary parts of the refractive index within the range of values from Di Biagio et al. (2019) (10% percentile).

[f]More absorptive SW refractive indies were calculated by varying the values of the imaginary parts of the refractive index within the range of values from Di Biagio et al. (2019) (90% percentile).

[g]More absorptive LW refractive indices were calculated by varying the values of the imaginary parts of the refractive index within the range of values from Di Biagio et al. (2017) (90% percentile).

[h]Less absorptive LW refractive indices were calculated by varying the values of the imaginary parts of the refractive index within the range of values from Di Biagio et al. (2017) (10% percentile).

**Table 2.** Summary of radiative effects estimated in this study.

| SW radiative effect | LW radiative effect | Difference |
|---|---|---|
| Less absorptive SW, coarser particle size, & aspherical shape | Coarser particle size & aspherical shape | E2 – E1 |
| Less absorptive SW & aspherical shape | Less absorptive LW & aspherical shape | E3 – E1 |
| Size-resolved dust abundance | Size-resolved dust abundance | E3 – E4 |
| Aspherical shape | Aspherical shape | E5 – E4 |
| Mineralogical variability in refractive index (more absorptive SW) | Mineralogical variability in refractive index (more absorptive LW) | E6 – E4 |
| Less absorptive SW (10% percentile) | More absorptive LW (90% percentile) | E7 – E4 |
| Less absorptive SW (10% percentile) | Less absorptive LW (10% percentile) | E8 – E4 |
| More absorptive SW (90% percentile) | More absorptive LW (90% percentile) | E9 – E4 |
| More absorptive SW (90% percentile) | Less absorptive LW (10% percentile) | E10 – E4 |

**Table 3.** Annually averages of dust load (Tg), mass extinction efficiency ($m^2 \cdot g^{-1}$), and $DAOD_{550}$ at each bin on a global scale. The size-resolved dust concentration and shape in IMPACT-Sphere-Mineral-V83 (E1) simulation was adjusted to DustCOMM in DustCOMM-Asphere-DB19-V83 (E2) simulation. At the same time, we maintained the consideration of asphericity on the gravitational velocity and kept the dust concentrations unaltered between IMPACT-Sphere-Mineral-V83 (E1) and IMPACT-Asphere-DB19-DB17 simulations (E3).

| Dust size bin | Dust load | | | Mass extinction efficiency | | | | $DAOD_{550}$ | | | |
|---|---|---|---|---|---|---|---|---|---|---|---|
| | E1 | E2 | DustCOMM | E1 | E2 | E3 | DustCOMM | E1 | E2 | E3 | DustCOMM |
| Bin 1[a] | 1.2 | 0.8 | 1.2 ± 0.7 | 2.11 | 3.41 | 3.33 | 3.06 | 0.0050 | 0.0055 | 0.0078 | 0.0070 |
| Bin 2 (1.26–2.5 µm) | 4.7 | 2.6 | 3.5 ± 2.1 | 0.73 | 1.25 | 1.21 | 1.22 | 0.0067 | 0.0064 | 0.0111 | 0.0084 |
| Bin 3 (2.5–5 µm) | 8.2 | 6.2 | 6.8 ± 3.8 | 0.37 | 0.59 | 0.57 | 0.57 | 0.0060 | 0.0071 | 0.0092 | 0.0077 |
| Bin 4 (5–20 µm) | 10.9 | 22.2 | 16.8 ± 9.0 | 0.23 | 0.24 | 0.29 | 0.19 | 0.0050 | 0.0104 | 0.0063 | 0.0063 |
| Sum of 4 bins | 25.0 | 31.8 | 28.4 ± 15.5 | 0.46 | 0.47 | 0.70 | 0.53 | 0.0227 | 0.0295 | 0.0345 | 0.0294 |

[a]Bin 1 in IMPACT-Sphere-Mineral-V83 (E1) is 0.1–1.26 µm, whereas bin1 in DustCOMM-Asphere-DB19-V83 (E2) and DustCOMM is 0.2–1.26 µm.

**Table 4.** Semi-observationally-based data set of clear-sky dust radiative effect efficiency at the surface and TOA.

| Number | Region name | Season | Region coordinates | Aerosol type selection | AOD data |
|---|---|---|---|---|---|
| R1[a] | Sahara Desert | Summer | 15°–30°N, 10°W–30°E | No selection | OMI-MISR |
| R2[b] | Tropical Atlantic | Summer | 15°–25°N, 45°–15°W | MODIS effective radius peaks 0.8–0.9 μm | MODIS |
| R3[c] | Tropical Atlantic | Summer | 10°–30°N, 45°–20°W | CALIOP dust and polluted dust | CERES-CALIPSO-CloudSat-MODIS |
| R4[d] | Atlantic Ocean | Summer | 0°–30°N, 60°–10°W | Dust detection based on $DAOD_{550}$ and fraction | MODIS |
| R5[e,f] | North Africa | Summer | 15°–35°N, 18°W–40°E | No selection | MISR[e] or SEVIRI[f] |
| R6[e,f] | West Africa | Summer | 16°–28°N, 16°–4°W | No selection | MISR[e] or SEVIRI[f] |
| R7[e,f] | Niger-Chad | Summer | 15°–20°N, 15°–22°E | No selection | MISR[e] or SEVIRI[f] |
| R8[e,f] | Sudan | Summer | 15°–22°N, 22°–36°E | No selection | MISR[e] or SEVIRI[f] |
| R9[e,f] | Egypt-Israel | Summer | 23°–32°N, 23°–35°E | No selection | MISR[e] or SEVIRI[f] |
| R10[e,f] | North Libya | Summer | 27°–33°N, 15°–25°E | No selection | MISR[e] or SEVIRI[f] |
| R11[e,f] | South Libya | Summer | 23°–27°N, 15°–25°E | No selection | MISR[e] or SEVIRI[f] |
| R12[g] | Mediterranean | Summer | 35.5°N, 12.6°E | Dust detection based on optical property | Ground-based measurements |
| R13[h] | Cape Verde | Summer | 16.7°N, 22.9°E | Dust detection based on brightness temperature | Ground-based measurements |
| R14[i] | China | Spring | 39°N, 101°E | Dust detection based on brightness temperature | Ground-based measurements |

[a]Yang et al. (2009). [b]Li et al. (2004). [c]Song et al. (2018). [d]Christopher and Jones (2007). [e]Zhang and Christopher (2003). [f]Brindley and Russell (2009). [g]Di Biagio et al. (2010). [h]Hansell et al. (2010). [i]Hansell et al. (2012).

**Table 5.** Annually averages of short-wave (SW) (W·m$^{-2}$), long-wave (LW) (W·m$^{-2}$), and net radiative effect (NET) (W·m$^{-2}$) at the surface, TOA, and atmospheric radiative heating on a global scale.

| Number | Data | Total dust SW | | Total dust LW | | Total dust NET | |
|---|---|---|---|---|---|---|---|
| | | TOA (surface)[a] | Atmosphere | TOA (surface)[a] | Atmosphere | TOA (surface)[a] | Atmosphere |
| E1 | IMPACT-Sphere-Mineral-V83 | −0.18 (−1.26) | 1.07 | +0.18 (0.66) | −0.48 | −0.00 (−0.60) | 0.59 |
| E2 | DustCOMM-Asphere-DB19-V83 | **−0.32 (−1.23)**[b] | **0.91**[b] | **+0.23 (1.00)**[b] | **−0.77**[b] | **−0.08 (−0.23)**[b] | **0.15**[b] |
| E3 | IMPACT-Asphere-DB19-DB17 | −0.49 (−1.35) | 0.86 | +0.12 (0.50) | −0.38 | −0.37 (−0.84) | 0.48 |
| E4 | DustCOMM-Asphere-DB19-DB17 | **−0.32 (−1.23)**[b] | **0.91**[b] | +0.12 (0.58) | −0.46 | −0.20 (−0.65) | 0.45 |
| E5 | DustCOMM-Sphere-DB19-DV17 | −0.28 (−0.90) | 0.62 | +0.08 (0.43) | −0.34 | −0.20 (−0.47) | 0.28 |
| E6 | DustCOMM-Asphere-Mineral-V83 | +0.02 (−1.61) | 1.63 | **+0.23 (1.00)**[b] | **−0.77**[b] | +0.25 (−0.62) | 0.87 |
| E7 | DustCOMM-Asphere-Less-More | −0.54 (−0.98) | 0.43 | +0.16 (0.76) | −0.60 | −0.38 (−0.22) | −0.16 |
| E8 | DustCOMM-Asphere-Less-Less | −0.54 (−0.98) | 0.43 | +0.06 (0.35) | −0.29 | −0.48 (−0.36) | 0.15 |
| E9 | DustCOMM-Asphere-More-More | −0.08 (−1.51) | 1.43 | +0.16 (0.76) | −0.60 | +0.09 (−0.75) | 0.84 |
| E10 | DustCOMM-Asphere-More-Less | −0.08 (−1.51) | 1.43 | +0.06 (0.35) | −0.29 | −0.01 (−1.16) | 1.15 |
| | DustCOMM (Adebiyi & Kok, 2020) | −0.59 to 0.17[c] | | +0.25 to 0.41[c] | | −0.27 to 0.14[c] | |
| M1 | Miller et al. (2004b) | −0.33 (−1.82) | 1.49 | +0.15 (0.18) | −0.03 | −0.18 (−1.64) | 1.46 |
| M2 | Tanaka et al. (2007) | −0.38 (−1.22) | 0.84 | +0.16 (0.57) | −0.41 | −0.22 (−0.65) | 0.43 |
| M3 | Yoshioka et al. (2007) | −0.92 (−1.59) | 0.67 | +0.31 (1.13) | −0.81 | −0.60 (−0.46) | −0.14 |
| M4 | Takemura et al. (2009) | −0.10 (−0.38) | 0.28 | +0.09 (0.18) | −0.09 | −0.01 (−0.20) | 0.19 |
| M5 | Albani et al. (2014) | −0.38 (−1.20) | 0.81 | +0.15 (0.64) | −0.49 | −0.23 (−0.56) | 0.33 |
| M6 | Colarco et al. (2014) | −0.32 (−1.25) | 0.93 | +0.05 (0.30) | −0.25 | −0.27 (−0.95) | 0.68 |
| M7 | Di Biagio et al. (2020) | −0.29 (−1.17)[d] | 0.88[d] | +0.23 (0.48)[d] | −0.26[d] | −0.06 (−0.69)[d] | 0.63[d] |
| M8 | Balkanski et al. (2021) | −0.14 (−1.42) | 1.28 | +0.12 (0.41) | −0.29 | −0.02 (−1.01) | 0.98 |

[a]The parentheses represent the RE at the surface. [b]The bold represents the combination of DB19 for SW and V83 for LW (i.e., DustCOMM-Asphere-DB19-V83). [c]98% confidential interval of DustCOMM data set is listed. [d]For a comparison with our estimates, sum of single mode simulations from Di Biagio et al. (2019) is listed.

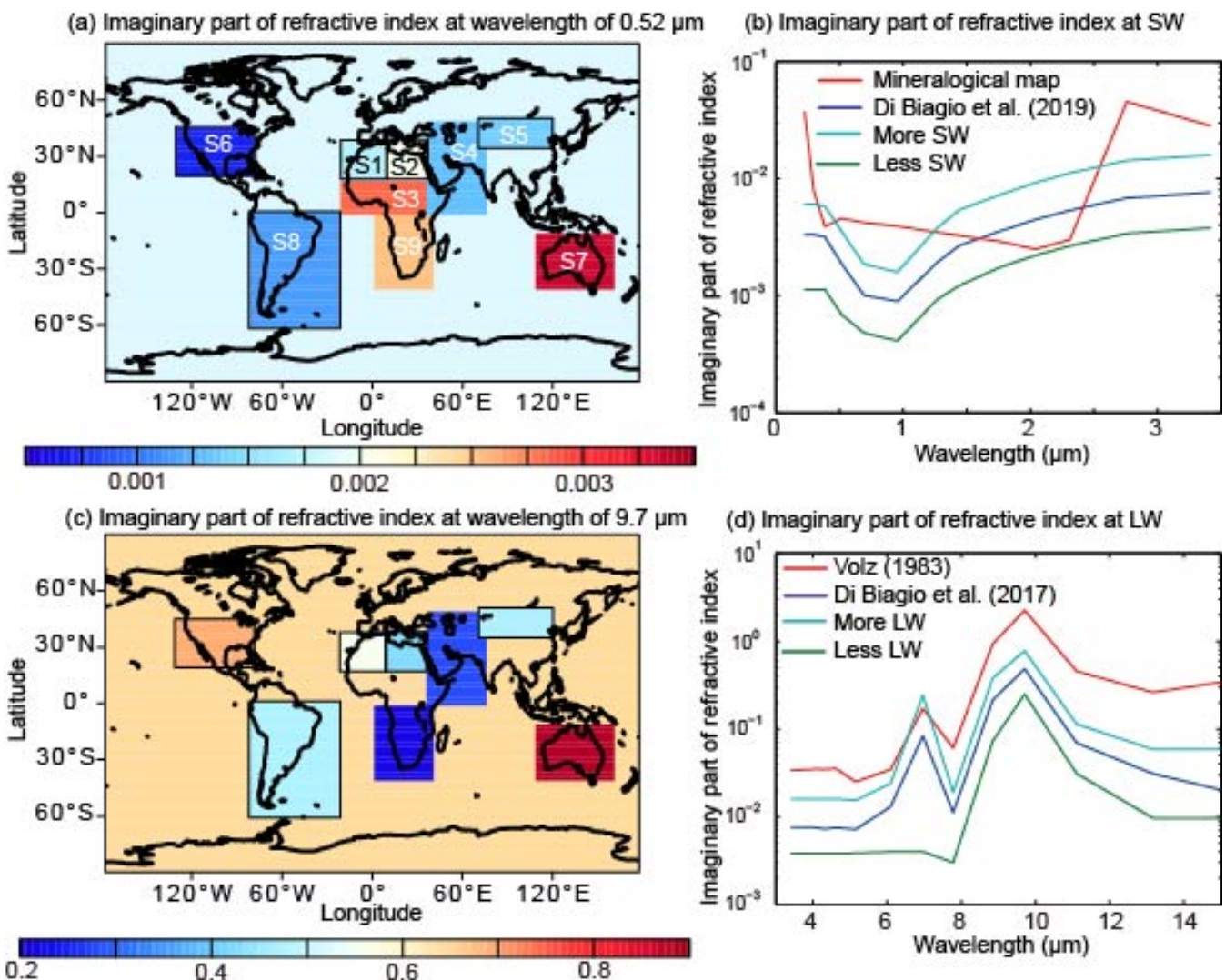

**Figure 1.**    Imaginary part of the refractive index at (a) 0.52 μm, (b) SW, (c) 9.7 μm, and (d) LW. The refractive index obtained from 19 samples was aggregated into 9 main source regions and the arithmetic mean was calculated for each source region (Di Biagio et al., 2017, 2019). The global mean is used for others. The coordinates of the nine source regions were: (S1) western North Africa (20°W – 7.5°E; 18°N – 37.5°N), (S2) eastern North Africa (7.5°E – 35°E; 18°N – 37.5°N), (S3) the Sahel (20°W – 35°E; 0°N – 18°N), (S4) Middle East / Central Asia (30°E – 70°E for 0°N – 35°N, and 30°E – 75°E for 35°N – 50°N), (S5) East Asia (70°E – 120°E; 35°N – 50°N), (S6) North America (130°W – 80°W; 20°N – 45°N), (S7) Australia (110°E – 160°E; 10°S – 40°S), (S8) South America (80°W – 20°W; 0°S – 60°S), and (S9) Southern Africa (0°E – 40°E; 0°S – 40°S).

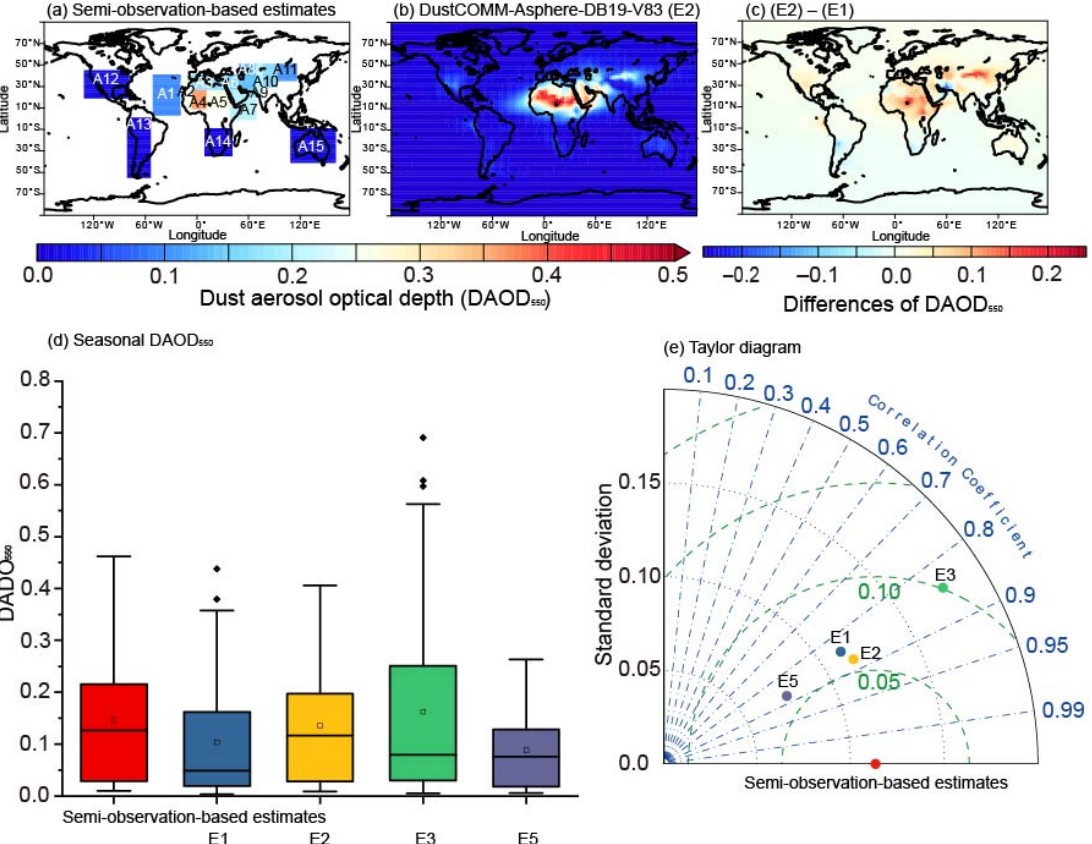

**Figure 2.** The model better reproduced semi-observationally-based data of DAOD$_{550}$ after adjusting the size-resolved dust load with DustCOMM and considering the dust asphericity. (a) semi-observationally-based estimates of the DAOD$_{550}$ were averaged over 2004–2008 (Ridley et al., 2016; Adebiyi et al., 2020). The annually averaged model results were shown for (b) DustCOMM-Asphere-DB19-V83 (E2) and (c) the differences between IMPACT-Sphere-Mineral-V83 (E1) and E2 simulations. (d) Comparison of seasonally averaged DAOD$_{550}$ for semi-observationally-based (SOB) data, E1, E2, IMPACT-Asphere-DB19-DB17 (E3), and DustCOMM-Sphere-DB19-DB17 (E5). The square symbol represents the mean. The solid line within the box mark shows the median. The boundaries of the box mark the 25th and 75th percentiles. The whiskers above and below the box indicate the 1.5 × interquartile range, and the points indicate the outside of the range. (e) Taylor diagram summarizing the statistics of the comparison against the seasonally averaged regional DAOD$_{550}$ for the different experiments. The horizontal axis shows the standard deviation of the data set or model prediction, the curved axis shows the correlation, and the green dashed lines denote the root-mean-squared errors between the semi-observationally-based data and the model predictions. As such, the distance between the semi-observationally-based data and the model predictions is a measure of the model's ability to reproduce the spatiotemporal variability in the semi-observationally-based data. The coordinates and the values of DAOD$_{550}$ at the 15 regions (marked in Fig. 2a) in summer were listed in Table S1. The comparison for other seasons was presented in Table S2.

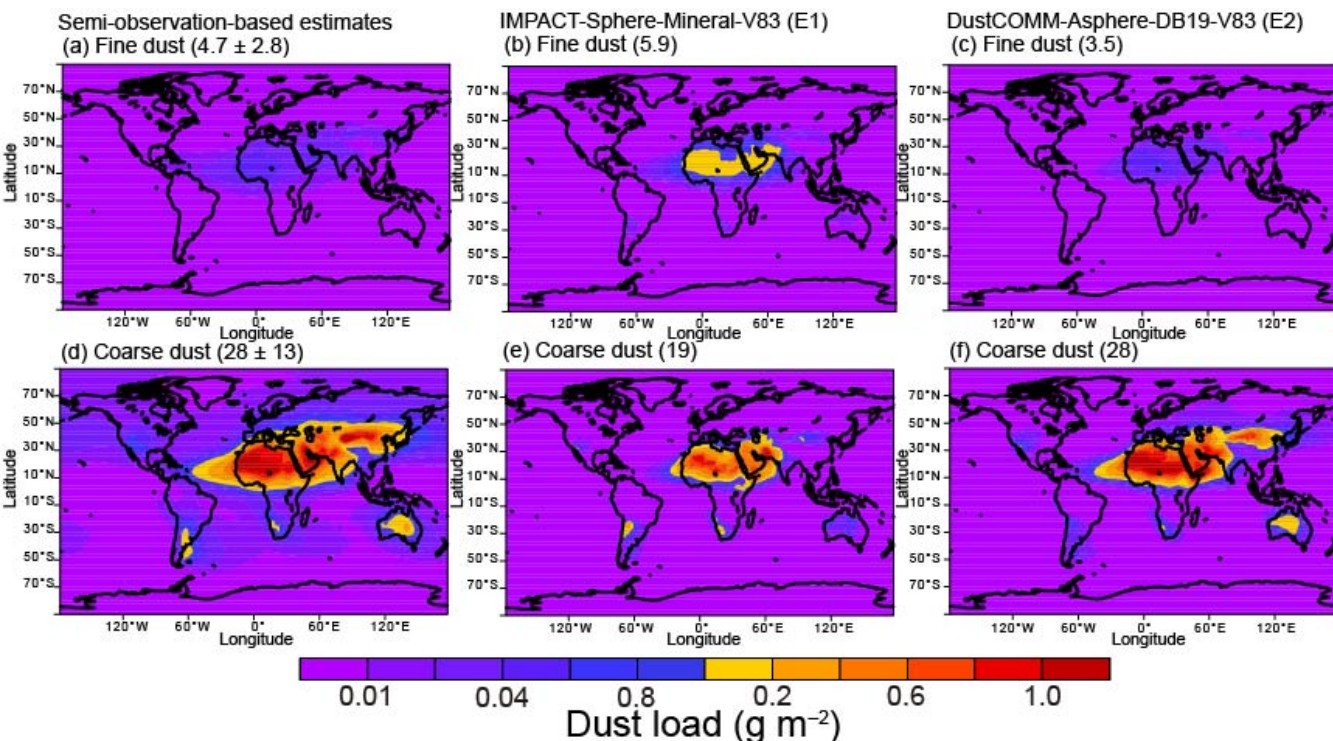

**Figure 3.**    Model-simulated dust loads at fine (smaller than 2.5 μm of diameter) and coarse size ranges (larger than 2.5 μm of diameter) before and after adjusting the size-resolved dust load with DustCOMM. Results were shown for (a) fine dust from DustCOMM, (b) fine dust from IMPACT-Sphere-Mineral-V83 (E1), (c) fine dust from DustCOMM-Asphere-DB19-V83 (E2), (d) coarse dust from DustCOMM, (e) coarse dust from E1, and (f) coarse dust from E2 simulations. The parentheses represented the global dust burden (Tg). The values of dust load at each bin were listed in Table 3.

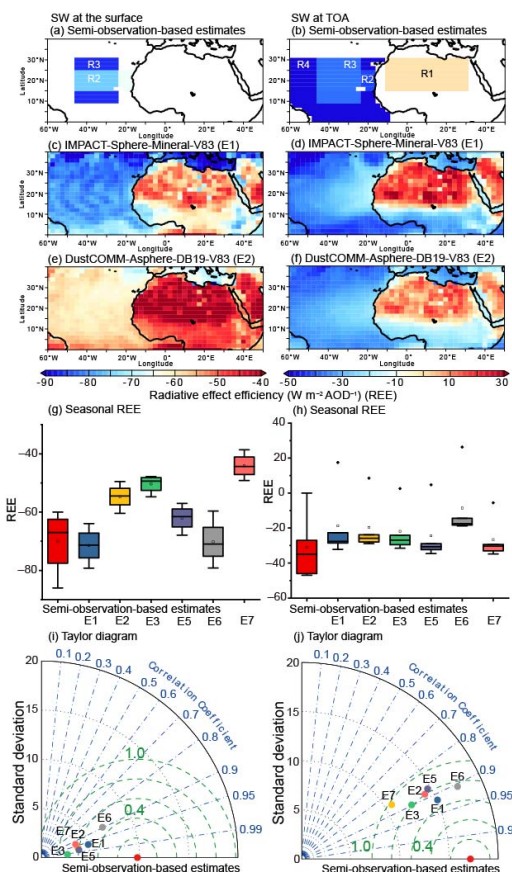

**Figure 4.** Dust clear-sky SW radiative effect efficiency (W·m$^{-2}$ DAOD$^{-1}$). Semi-observationally-based data at (a) the surface and (b) TOA were based on satellite observations (Yang et al. 2009; Li et al., 2004: Song et al., 2018; Christopher and Jones, 2007). The model results were shown for (c) and (d) IMPACT-Sphere-Mineral-V83 (E1), and (e) and (f) DustCOMM-Asphere-DB19-V83 (E2) simulations at the surface and TOA, respectively. Comparison of seasonally averaged SW radiative effect efficiency for semi-observationally-based (SOB) data and the different experiments at (g) the surface and (h) TOA. The square symbol represents the mean. The solid line within the box mark shows the median. The boundaries of the box mark the 25th and 75th percentiles. The whiskers above and below the box indicate the 1.5 × interquartile range, and the points indicate the outside of the range. Taylor diagram summarizing the statistics of the comparison against the seasonally averaged regional SW radiative effect efficiency for the different experiments at (i) the surface and (j) TOA. The horizontal axis shows the standard deviation of the data set or model prediction, the curved axis shows the correlation, and the green dashed lines denote the root-mean-squared errors between the semi-observationally-based data and the model predictions. As such, the distance between the semi-observationally-based data and the model predictions is a measure of the model's ability to reproduce the spatiotemporal variability in the semi-observationally-based data. The regionally averaged values were listed in Tables S3 and S4 at the surface and TOA, respectively.

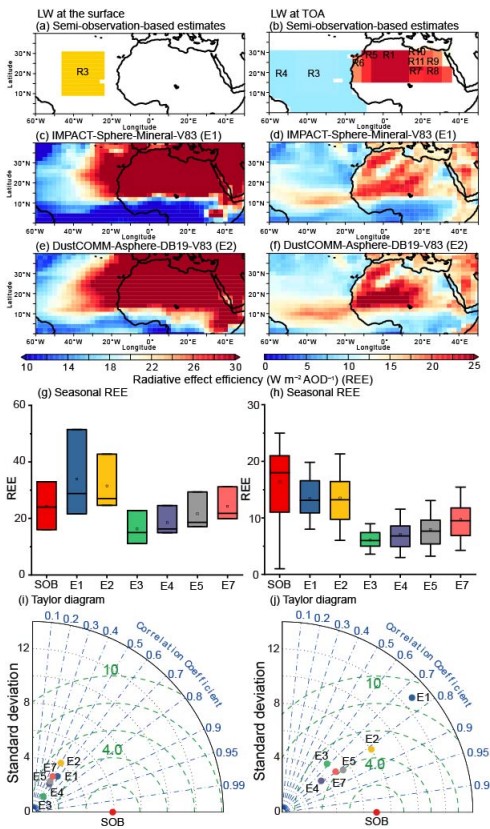

**Figure 5.** Dust clear-sky LW radiative effect efficiency (W·m$^{-2}$ DAOD$^{-1}$). Semi-observationally-based estimates at (a) surface and (b) TOA were based on satellite observations (Song et al., 2018; Christopher and Jones, 2007; Zhang and Christopher, 2003; Brindley and Russell, 2009; Yang et al., 2009). The model results were shown for (c) and (d) IMPACT-Sphere-Mineral-V83 (E1), and (e) and (f) DustCOMM-Asphere-DB19-V83 (E2) simulations at the surface and TOA, respectively. Comparison of seasonally averaged LW radiative effect efficiency for semi-observationally-based (SOB) data and the different experiments at (g) the surface and (h) TOA. The square symbol represents the mean. The solid line within the box mark shows the median. The boundaries of the box mark the 25th and 75th percentiles. The whiskers above and below the box indicate the 1.5 × interquartile range, and the points indicate the outside of the range. Taylor diagram summarizing the statistics of the comparison against the seasonally averaged regional SW radiative effect efficiency for the different experiments at (i) the surface and (j) TOA. The horizontal axis shows the standard deviation of the data set or model prediction, the curved axis shows the correlation, and the green dashed lines denote the root-mean-squared errors between the semi-observationally-based data and the model predictions. As such, the distance between the semi-observationally-based data and the model predictions is a measure of the model's ability to reproduce the spatiotemporal variability in the semi-observationally-based data. The regionally averaged values were listed in Tables S5 and S6 at the surface and TOA, respectively.

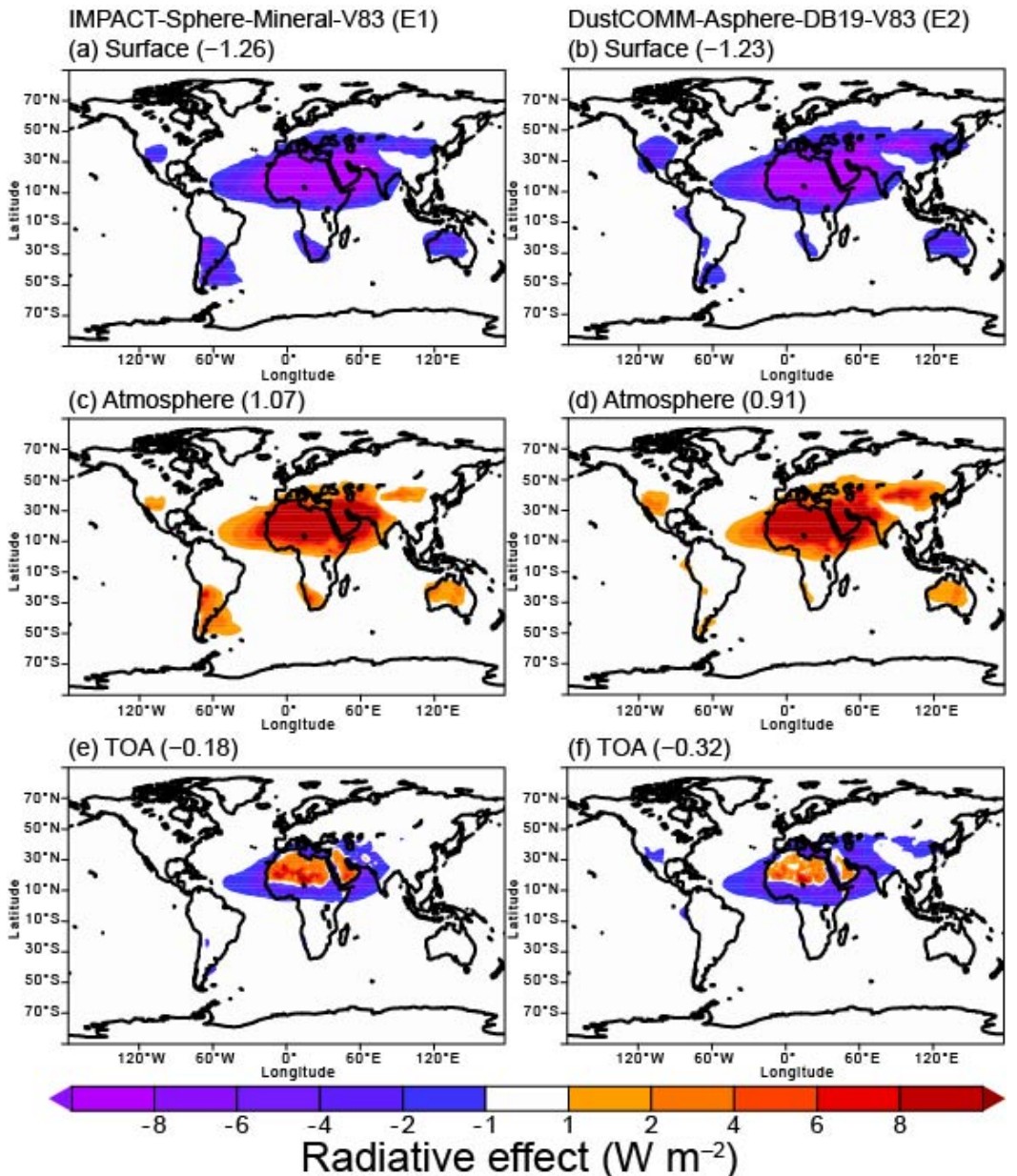

65

**Figure 6.**    Dust SW radiative effect (W·m$^{-2}$) and radiative heating of the atmosphere (i.e., the subtraction of radiative effects from TOA to the surface in unit of W·m$^{-2}$). The model results were shown for the simulations for (a) IMPACT-Sphere-Mineral-V83 (E1) at the surface, (b) DustCOMM-Asphere-DB19-V83 (E2) at the surface, (c) E1 in atmospheric column, (d) E2 in atmospheric column, (e) E1 at TOA, and (f) E2 simulations at TOA. The numbers in parentheses represented the global

70    mean.

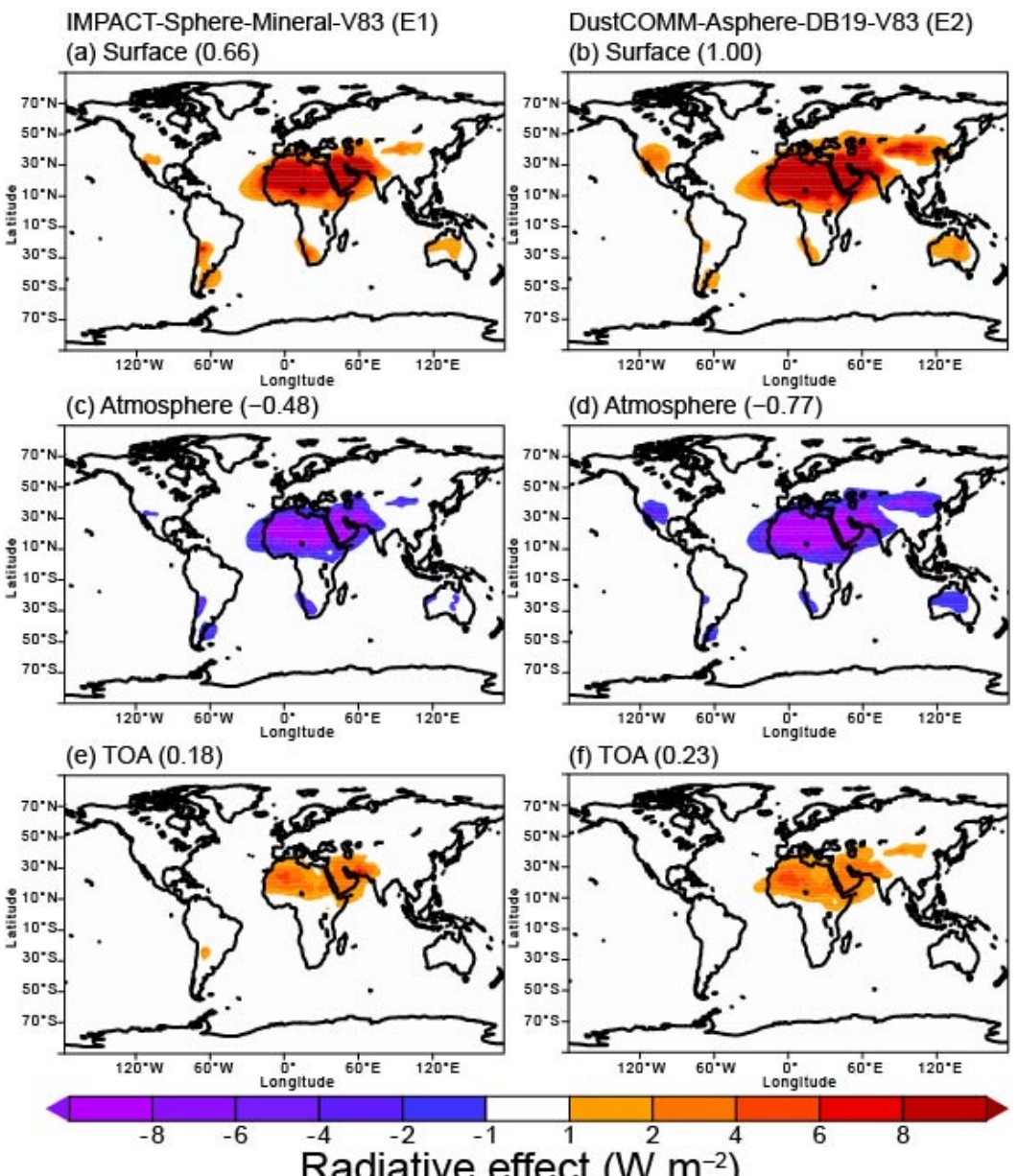

**Figure 7.**  Dust LW radiative effect (W·m$^{-2}$) and radiative heating of the atmosphere (i.e., the subtraction of radiative effects from TOA to the surface in unit of W·m$^{-2}$). The model results were shown for the simulations for (a) IMPACT-Sphere-Mineral-V83 (E1) at the surface, (b) DustCOMM-Asphere-DB19-V83 (E2) at the surface, (c) E1 in atmospheric column, (d) E2 in atmospheric column, (e) E1 at TOA, and (f) E2 simulations at TOA. The numbers in parentheses represented the global mean.

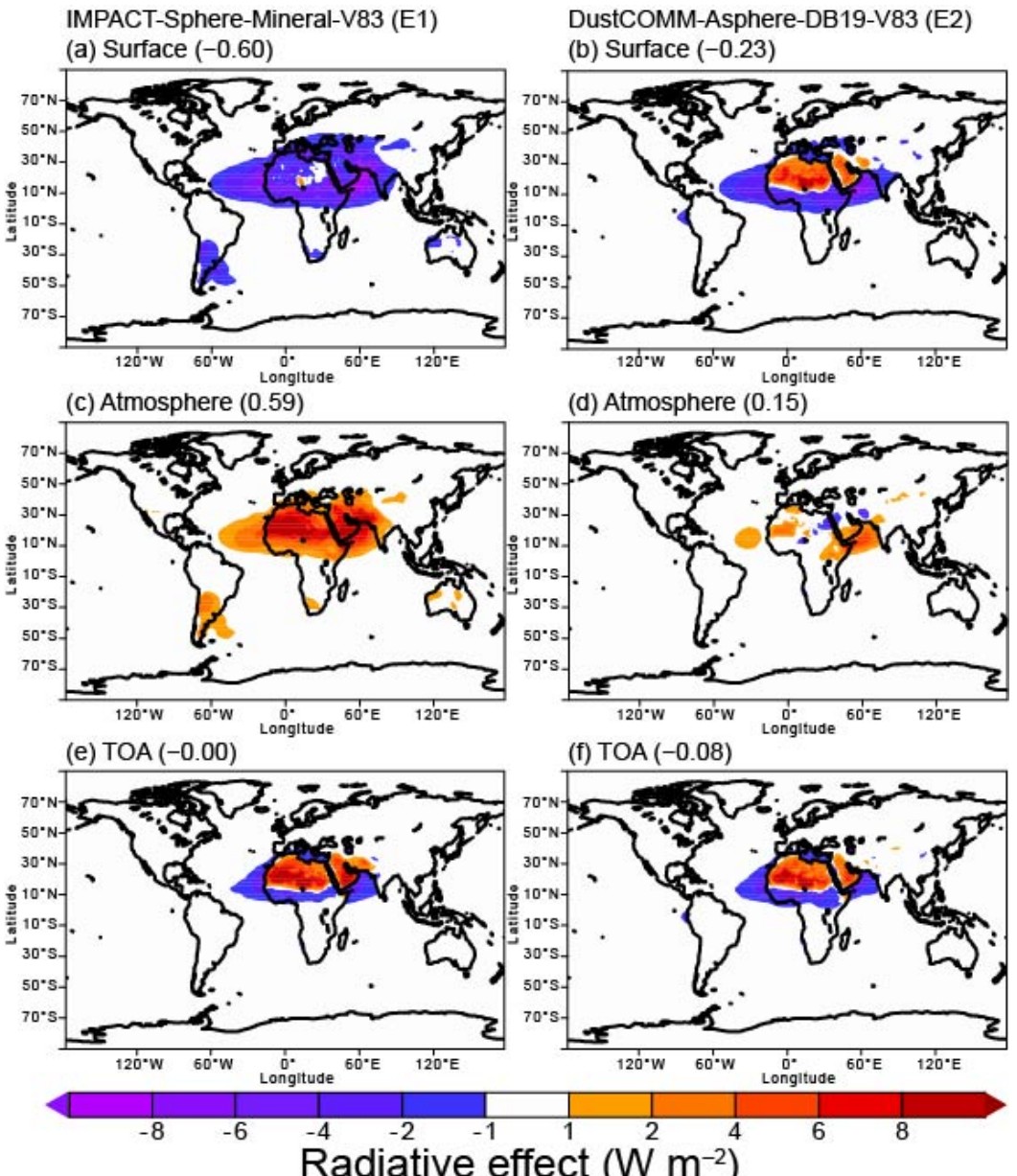

**Figure 8.** Dust net radiative effect ($W·m^{-2}$) and radiative heating of the atmosphere (i.e., the subtraction of radiative effects from TOA to the surface in unit of $W·m^{-2}$). The model results were shown for the simulations for (a) IMPACT-Sphere-Mineral-V83 (E1) at the surface, (b) DustCOMM-Asphere-DB19-V83 (E2) at the surface, (c) E1 in atmospheric column, (d) E2 in atmospheric column, (e) E1 at TOA, and (f) E2 simulations at TOA. The numbers in parentheses represented the global mean.

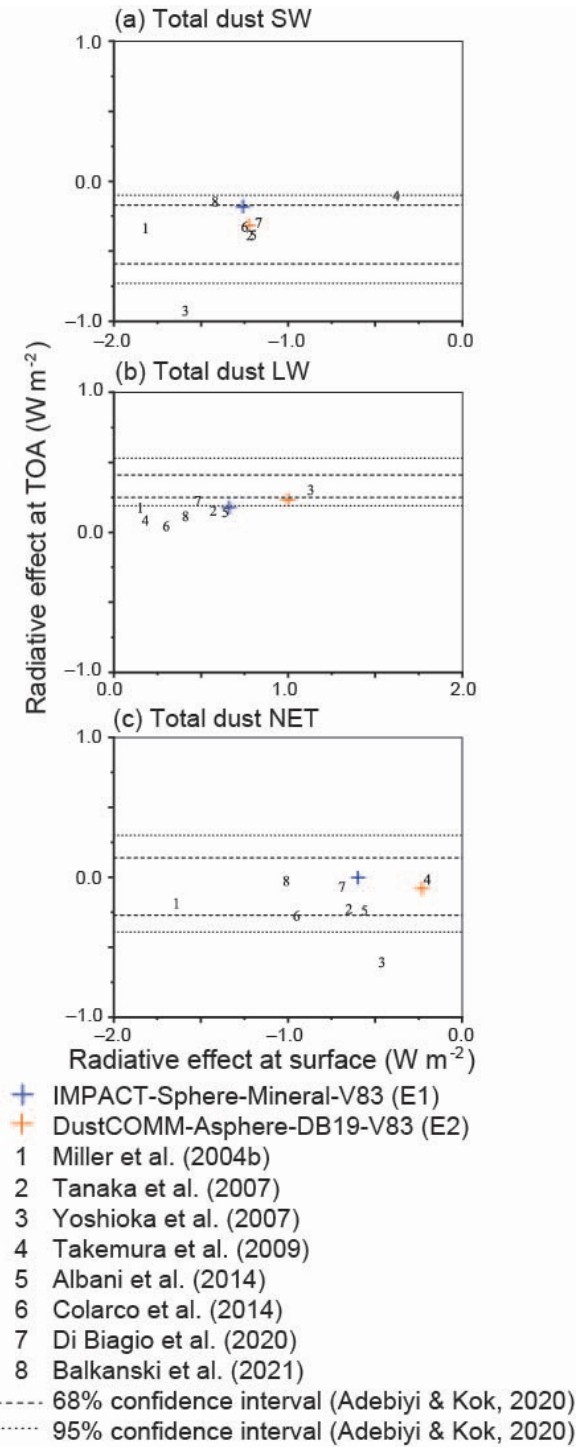

**Figure 9.** Variability of dust radiative effect (W·m⁻²) in different model simulations at the surface and TOA for (a) total dust SW, (b) total dust LW, and (c) total dust NET. The annually averaged values were listed in Table 5.

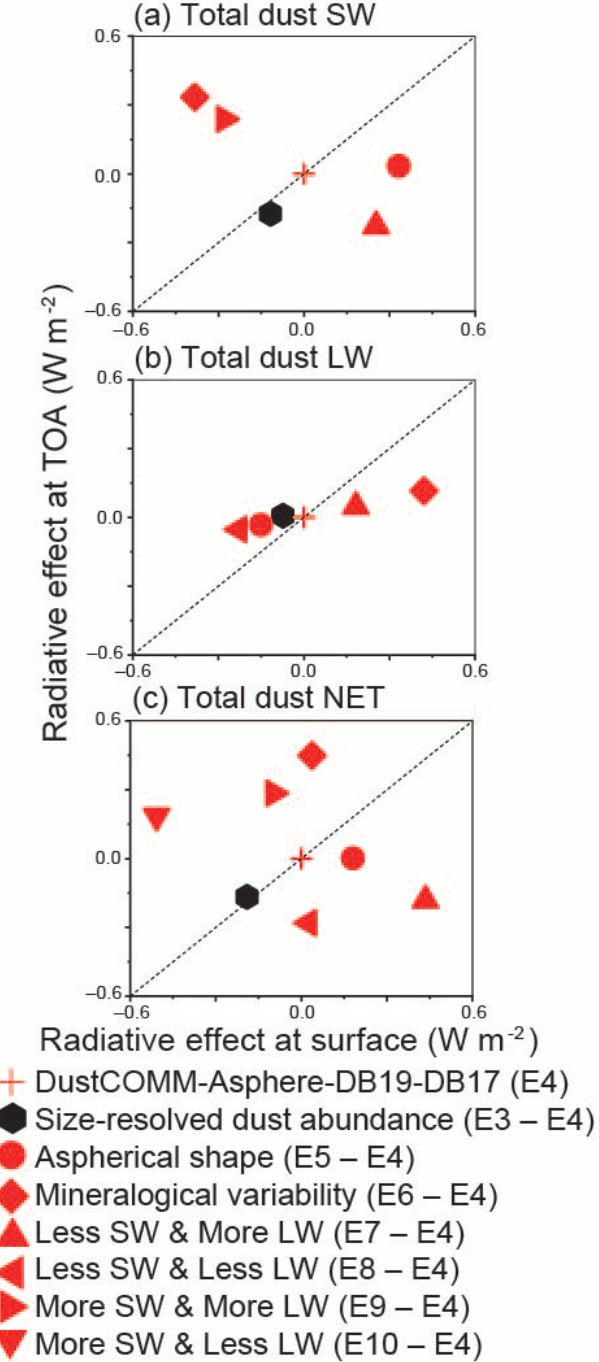

**Figure 10.** Radiative effect ($W \cdot m^{-2}$) of mineral dust due to various aerosol absorptivity at the surface and TOA for (a) total dust SW, (b) total dust LW, and (c) total dust NET. The annually averaged values were listed in Table 5. The dashed line

90  represented a 1 : 1 correspondence and corresponded to no change in radiative heating within the atmosphere.