# Peer review of "Less atmospheric radiative heating by dust due to the synergy of"

_Atmospheric Chemistry and Physics, 2021_

## Author Response (AR1)

Our responses to the comments from Reviewer 1 are included in the followings.

Reviewer #1

**Comments:**

The present work aims at investigating the direct radiative effect of mineral dust aerosols and in particular it focuses on the effect of dust asphericity. This analysis takes advantage of a strong background of work performed in past years by the authors of the manuscript (work on size distribution and inclusion of non-spherical spectral optical properties calculations, Kok et al. 2017 Nat geosci ; DustCOMM dataset creation, Adebiyi et al., ACP, 2020 ; study of the asphericity of dust and inclusion of the effect on reducing gravitational settling, Huang et al., GRL, 2020) to realise new model simulations with the IMPACT model coupled with the RRTMG radiation code. The IMPACT model has been improved in this study compared to its default configuration by accounting mainly of : a better soil moisture dataset from satellite observations to improve dust emissions, to include the effect of asphericty on gravitational settling velocity, by integrating with the RRTMG radiative code, and by using the DUSTCOMM dataset to constrain simulations. The paper provides a series of sensitivity simulations varying mainly the size distribution and the spectral optical properties of mineral dust aerosols and assuming or not the effect of asphericity in the simulations. The results are compared to field observations of the dust DREE (direct radiative effect efficiency, Wm-2 AOD-1) to identify the simulation configuration that best reproduces field measured dust perturbations at the surface and at TOA. The main conclusion of the paper is that improving the simulation scheme (improved vs default simulations) does not significantly changes the TOA global annual net DRE, conversely both the surface cooling and the atmospheric heating are strongly reduced assuming coarse aspherical dusts.

The paper is potentially a nice contribution for the scientific community. It provides interesting insight on dust aerosol science and contributes to further advance in the modeling of the dust cycle and its direct effects. Despite, I find that the paper suffers from an unclear identification of the objectives and a poor contextualization compared to the recent literature. As well, the presentation of the modelling simulations and the description and discussion of the results should be improved. In the current form I find the paper a bit difficult to read and I suggest that some major revisions are applied for improving the organisation and the presentation/discussion of the results. I compiled some comments below.

**Response:**

We would like to thank the reviewer for his or her constructive comments which helped us to improve the readability of our manuscript substantially. We thoroughly revised the manuscript following suggestions made by the reviewer.

**General comments**

**Comment 1:**

One of my problem, expecially at the very first reading, was to understand the main objective of the paper. I had the feeling by reading the title that asphericity was the main topic, then in the

introduction it is discussed that asphericity is not so important (see lines 65 to 74) and then the paper introduces many simulations testing the dust DRE sensitivity and many test studies. The overall introduction and description of the work should be more incisive and clear in the objectives and scientific questions to test.

**Response:**

Thank you for your suggestions. To elucidate the importance of asphericity is one of our main topics. In the revised paper, we emphasized the importance of asphericity for the calculation of dust aerosol spectral optical depth and radiative effect. To do so, we presented an additional simulation result (Experiment 5) to elucidate the asphericity effects. We revised the title, "Less atmospheric radiative heating by dust due to the synergy of coarser size and aspherical shape", and paragraphs in the introduction on p.3, l.68 and l.75:

"Second, previous studies have shown that the SW radiative effect of dust asphericity on climate simulations is minor on a global scale, partly because the larger DAOD is compensated for by the larger asymmetry parameter of aspherical dust, which reduces the amount of radiation scattered backward to space (Räisänen et al., 2013; Colarco et al., 2014)."

"However, the assumption of spherical shape in models leads to a substantial underestimation of the extinction efficiency and thus DAOD near the strong source regions, mainly because the assumption of sphericity causes an underestimation of the surface-to-volume ratio compared to aspherical dust (Kok et al., 2017, 2021; Hoshyaripour et al., 2019; Tuccella et al., 2020). Radiative effect efficiency is often used for the evaluation of the models and is defined as the gradient of a linear least squares fit applied to AOD and dust radiative effect at each two-dimensional (2-D) grid box (W·m$^{-2}$ AOD$^{-1}$). Thus, the estimates of the dust radiative effect efficiency could be biased, in part, due to large uncertainties associated with the spherical assumption on AOD retrieval (Zhou et al., 2020)."

**Comment 2:**

Following the previous comment, I have also found a little bit tricky to follow/understand the scope of the many simulations performed despite the synthesis effort in Table 1 and 2. There are many things and concepts in these simulations and a non-expert reader could be lost. Probably be clearer in describing the strategy?

**Response:**

In revised paper, we changed the structure of subsection and added outline of the methodology used to obtain the sensitivity simulations of dust radiative effects: (1) the size-resolved abundance, (2) particle shape, and (3) mineralogical variability on p.4, l.118:

"In section 2.3, we describe the DustCOMM data set used to adjust (1) size-resolved abundance of dust concentration. In section 2.4, we describe the adjustment factor of (2) particle shape for spectral optical properties. In section 2.5, we describe differences in spectral refractive indices due to (3) different mineralogical compositions for the radiative flux calculation."

**Comment 3:**

The paper refers too much to the supporting information, in particular in the Results section. While on one side I appreciate the effort of synthesis, I feel that it is quite difficult to follow the reasoning when being obliged to go back and forth from the main paper to SI. I suggest the authors to consider revising their strategy of presentation of the results in order to help the reader following the reasoning.

**Response:**

In the revised paper, we showed the evaluation of the various model experiments against semi-observationally-based estimates in box plots and Taylor diagrams (Taylor, 2001) in the main paper to provide a concise statistical summary of the bias, correlation coefficient, root mean square errors, and the ratio of standard deviation on p.11, l.315. Accordingly, we deleted supplementary figures.

"We compared our model estimates of $DAOD_{550}$ against semi-observationally-based data in box plots and Taylor diagrams (Taylor, 2001) for the evaluation of the various model experiments against semi-observationally-based estimates (Ridley et al., 2016; Adebiyi et al., 2020) to provide a concise statistical summary of the bias, correlation coefficient, root mean square errors, and the ratio of standard deviation (Fig. 2, Tables S1 and S2)."

**Comment 4:**

Referring to lines 51-53 « On the other hand, model errors due to the underestimated coarse dust load and corresponding warming might be compensated for in models by using a refractive index that is too absorbing (Di Biagio et al., 2019), and which depends on the mineral composition of the dust ». Isn't it the same case here based on your results? Here the size distribution is cut at 20µm despite field evidences that larger particles are efficiently retained during transport (see FENNEC or SALTRACE results) and the best agreement with observations is found then when a stronger absorption is assumed in particular in the LW range, where the contribution of the coarse dust component is more critical. Is this result just due to the missing coarse size in the model above 20 µm? I would expect such a discussion in the paper (I noticed a mention to this at the very end of the conclusions, but the issue is argued to be related only to « Godzilla » type events and not relevant elsewhere. Is this really true or this aspect deserve more discussion?)

**Response:**

We thank the reviewer for this insightful comment. Song et al. (2018) found that the combination of Fennec dust particle size distribution and OPAC-LW, which was originally taken from Volz (1983), yielded the best simulation of the dust LW radiative effect in comparison with the satellite flux observations (i.e., CERES OLR), compared to the Di Biagio et al. LW refractive index. Marine sediment traps suggest that giant particles are dominated by platy mica and rounded quartz particles (van der Does et al., 2016). Indeed, the dust sample for V83 was collected from rainwater after strong wind by Volz (1983). Thus, mineral composition of the giant particles could be different from the aerosol samples generated from soils in the laboratory by Di Biagio et al. (2017), which may reflect less absorbing LW refractive index of DB17 than V83, as Di Biagio et al. (2017) found a linear relationship between the magnitude of the imaginary refractive index at 7.0, 9.2,

and 11.4 μm and the mass concentration of calcite and quartz absorbing at these wavelengths. In revised paper, we replaced Experiment 1 with the same LW refractive index of Volz (1983) as in Experiment 2, and presented additional simulation results (Experiments 8, 9, and 10) to elucidate the sensitivity of radiative effect to refractive index.

Di Biagio et al. (2020) mentioned, "the key role of particles larger than 20 μm, however, does not only rely on their direct contribution to the DRE but mostly on the fact that their inclusion reduces the contribution by smaller (cooling) particles to the global dust cycle". We revised the sentence on p.2, l.43:

"The model errors in dust size distribution and particle shape can lead to an overestimate of fine dust load after the dust emissions in the models are scaled to match observed dust aerosol optical depth at 550 nm (DAOD$_{550}$). The corresponding overestimate of SW cooling might be compensated for in models by using a refractive index that is too absorbing (Di Biagio et al., 2019, 2020), which depends on the mineral composition of the dust."

The coarse size in the model above 20 μm deserves more discussion. The combination of less LW absorption and coarser size can be examined from revised Fig. 10b. The dust size beyond 20 μm might be partly compensated for in our model by using a refractive index that is more LW absorbing. Thus, we added the comparison with other modeling studies which considered the dust size beyond 20 μm (Di Biagio et al., 2020) to Table 5 and revised Fig. 9 on p.14, l.409:

"A relatively good agreement of net RE by dust at TOA with Di Biagio et al. (2020) (–0.06 W·m$^{-2}$) could be obtained from both the IMPACT-Sphere-Mineral-V83 (E1) and DustCOMM-Asphere-DB19-V83 (E2) simulations (Fig. 9 and Table 5). On the other hand, our modeled dust net RE at the surface from DustCOMM-Asphere-DB19-V83 (E2) was much larger than Di Biagio et al. (2020) (–0.63 W·m$^{-2}$) and IMPACT-Sphere-Mineral-V83 (E1). The synergy of coarser size and aspherical dust could contribute to the less surface warming of the DustCOMM-Asphere-DB19-V83 (E2). At the same time, the more absorptive LW dust refractive index (V83) than DB17 (red diamond in Fig. 10b) could also contribute to the less surface warming, which might be partially compensated for in our model by the omission of dust with diameters in excess of 20 μm. Consequently, our estimate of atmospheric radiative heating by dust from DustCOMM-Asphere-DB19-V83 (E2) was lower than Di Biagio et al. (2020) (0.63 W·m$^{-2}$) and IMPACT-Sphere-Mineral-V83 (E1)."

We added the following sentences to conclusions on p.15, l.456:

"Moreover, such large particles can be transported to higher altitudes and longer distances than the model prediction. The higher the dust layer resides, the larger the dust LW RE at TOA is estimated under the clear-sky conditions (Liao and Seinfeld, 1998). Marine sediment traps, which are located underneath the main Saharan dust plume in the Atlantic Ocean, suggest that giant particles are dominated by platy mica and rounded quartz particles (van der Does et al., 2016). Thus, mineral composition of the giant particles could be different from the aerosol samples generated from soils in the laboratory by Di Biagio et al. (2017), which may reflect less absorbing LW refractive index of DB17 than V83. Indeed, the dust sample was collected for V83 from rainwater after strong wind. On the other hand, the contribution of the LW scattering might be underestimated in the models,

as Di Biagio et al. (2020) noted that the adjustment factor was estimated for dust of diameter less than 10 μm and thus might be a lower approximation of the LW scattering by coarse dust. Therefore, a better understanding of the effect of such large particles beyond 20 μm and mineralogical composition on radiation balance remains a topic of active research, given their potential to amplify the warming of the climate system."

This was also reflected on introduction, p.3, l.91:

"However, the speciation of dust into its mineral components inherently comprises uncertainties on soil mineralogy, mineral content in size-segregated dust particles, and refractive index of mineral, partly due to the differences in prescribed parameters such as the particle size."

**Specific comments**

**Comment 1:**

Abstract : the concept of default and improved simulation si given, but I am not sure it is fully clear at this stage. I would remove this nomenclature from the abstract text.

**Response:**

The default and improved was rephrased by before and after the adjustment, as was suggested.

**Comment 2:**

Line 23 : please specify the temporal/spatial scale of these estimated effects (global annual I guess)

**Response:**

This was done, as was suggested.

**Comment 3:**

Line 27 : I would say « warming of the Earth-atmosphere system by trapping incident and outgoing radiation »

**Response:**

This was done, as was suggested.

**Comment 4:**

Line 27 : what do you mean with « climate feedback ». Is this the correct term here ?

**Response:**

We added following sentence before the climate feedback on p.1, l.34:
"Radiative effect by dust aerosols perturbs surface temperature, wind speed, rainfall, and vegetation cover, which may induce feedback on dust emissions (Perlwitz et al. 2001; Miller et al., 2004a; Colarco et al., 2014)."

**Comment 5:**

Lines 54-64 : probably a word here also on the minerals affecting LW absorption would be good

**Response:**

This was added, as was suggested on p.3, l.88:

"The dust complex refractive index in the LW also depends on the particle mineralogical composition (Sokolik et al., 1998). Di Biagio et al. (2017) found a linear relationship between the magnitude of the imaginary refractive index at 7.0, 9.2, and 11.4 μm and the mass concentration of calcite and quartz absorbing at these wavelengths."

**Comment 6:**

Lines 65-74 : I have to admit that I am quite confused here. The scope of the paper is to include asphericity effects but here I understand that it is not a big issue for global modelling. Please be more clear and focused on the objective and contours of the work.

**Response:**

This was revised, as was suggested in general comment 1.

**Comment 7:**

Lines 80-84 : it is quite unusual to draw the conclusions or the results in the introduction of the paper

**Response:**

This was removed, as was suggested.

**Comment 8:**

Line 91 : what is the forward model referred here ?

**Response:**

This is the IMPACT model simulation before the adjustment. The "forward" was rephrased by "IMPACT" in revised paper.

**Comment 9:**

Lines 104-112 : could be possible to give a bit more information or reference to the model capacity in reproducing dust mineralogy reasonably ?

**Response:**

The references of recent review papers including multi-model evaluations were added, as was suggested. We added the sentence on p.5, l.148:

"In recent review papers, multi-model evaluations of aerosol iron concentrations and their solubilities have been comprehensively summarized on global and regional scales (Myriokefalitakis et al., 2018; Ito et al., 2021)."

**Comment 10:**

Lines 123-128 : even by accounting for an « optimized » asphericity factor for dust in the model the lifetime of the coarsest bin is too low compared to pas literature. What is the impact of this on the results and overall sensitivity ? I guess this is a crucial point.

**Response:**

This is the one of our main topics and the reason why we adjusted the size-resolved abundance of dust concentration. The impact of this on the results and overall sensitivity was evaluated by "Experiment 3 – Experiment 4", as was summarized in Table 2. To elucidate this, we added the following sentence on p.6, l.166:

"The impact of this underestimate of atmospheric lifetime is explored using the DustCOMM data set, as was summarized in Table 2 (E3 – E4)."

**Comment 11:**

Line 134 : given that the focus of the paper is aspherity effects I am quite surprised to see that Mie theory is used here. However is then in section 2.4 that it is explained what it is done. Not sure the two things should not be merged together.

**Response:**

We presented the additional simulation result (Experiment 5) to show the asphericity effects. To elucidate this, we added the following sentence on p.6, l.177:

"The impact of this spherical assumption is explored using aspherical factor, as was summarized in Table 2 (E5 – E4)."

**Comment 12:**

Section 2.3 : I have to say that it remains a bit unclear how the dustcomm database is used in practice until sect 2.4. This sect 2.3 is a bit confusing to me

**Response:**

This was revised, as was suggested in general comment 1 and specific comment 10.

**Comment 13:**

Section 3.3 : should be taken in mind and reminded to the reader that the comparison in the LW range is more tricky than in the SW since there is a stronger dependence of the DRE on the vertical profile of dust, temperature and water vapour profiles therefore affecting the measured and modelled comparison

**Response:**

This was added, as was suggested on p.12, l.360:

"The dust LW radiative effect efficiency depends strongly on the vertical profile of dust concentration, temperature, and water vapor, which would affect the comparison due to a high variability in these factors (section 2.6)."

**Comment 14:**

Section 4 and throughout the text : pay attention to refer to « spectral optical properties » when related to both the SW and LW spectra

**Response:**

We referred to spectral optical properties when related to both the SW and LW spectra, as was suggested.

**References**

[revised manuscript text omitted]

Our responses to the comments from Reviewer 2 are included in the followings.

Reviewer #2

**Comments:**

What is presented here is a series of sensitivity analyses based on a global model simulation of atmospheric dust. The radiative effect of dust aerosols is computed under a number of different configurations, testing on different dust refractive index assumptions in shortwave and long wave and different particle size and shape assumptions. Sensitivity of the resulting radiative forcing efficiencies is demonstrated. The main conclusion of the paper is well stated in the title and in the abstract, introduction, and conclusions. The best simulation results from an adjustment of the default simulation toward the 3d, seasonal, and size constrained dust distributions from the DustCOMM database, non-spherical dust, and the combined lesser shortwave absorption in Di Biagio (2019) and the long wave assumptions of refractive index from the older Volz et al (1983) database.

The paper is compact and short in text, but there is a tremendous amount of information contained in 11 tables and 14 figures. The supplementary material (especially the tables) is so necessary I suggest it just be folded into the paper proper. It is actually a confusing and frustrating paper to navigate in how it is presently laid out and I think much to the detriment of the work, which could be significant if the presentation can be cleared up. I found this a difficult paper to read through, partly in need too reference several different figures and tables to get a point out and partly in some confusing choices I think the authors make in how to set this up. It needs more work to clear some of this up. I don't know how one could possibly read this on a computer or tablet (you'd need multiple copies open to see all the bits) and even printing it out it was a hard slog to keep it all organized.

I am not convinced of the central conclusion. In particular, Figure 4 and Table S3 seem not to justify the "Improved" simulation as better than "Default" for surface SW forcing efficiency, maybe "Coarse-mineral" is the best agreement at the surface though clearly worse at TOA. The various scatter plots do not seem to much different in correlation or even RMSE (e.g., Figure S2). It's a little hard to untangle though how this optimized solution was arrived at.

**Response:**

We would like to thank the reviewer for his or her constructive comments which helped us to improve the readability of our manuscript. We thoroughly revised the manuscript following suggestions made by the reviewer. To clarify our storyline, we added the background and guideline to abstract on p.1, l.10, section 1 on p.2, l.43, section 2 on p.4, l.118, and section 3 on p.10, l.303.

[revised manuscript text omitted]

**General comments**

**Comment 1:**

First, I suggest the distinction of "fine" and "coarse" in the simulation titles is a little confusing at the outset, as those terms are also used in the paper to refer to particular sections of the particle size distribution. "Fine-global" might be titled "Default-aspherical" or just "aspherical" which makes the perturbation clear. I suppose "Coarse" is then clear enough, or else "DustCOMM" to discriminate the size and loading adjustment. Anyway, lines 90 - 95 don't really map to table 1 well ("denoted as 'fine'" — well, only one experiment is actually denoted as "fine", and "denoted as 'coarse'" evidently includes "Improved" which is not denoted as "coarse"). Please clean this up.

**Response:**

Thank you for your suggestions. We rephrased "Fine" and "Coarse" by "IMPACT" and "DustCOMM", respectively, as was suggested. We revised the terms in Table 1 and on p.4, l.106:

"Two experiments used the dust concentrations calculated from the IMPACT model with the finer dust size (denoted as "IMPACT"). Subsequently, the simulated dust concentration and the size distribution were adjusted to the semi-observationally-based concentrations (Adebiyi and Kok, 2020) in the other eight experiments with the coarser dust size (denoted as "DustCOMM")."

**Comment 2:**

Second, I'm confused about how many simulations were actually run with IMPACT. Since I take that asphericity is included in the dust gravitational settling in the Default run then is there only a single simulation run and then everything else is handled in offline calculations performed with different look up tables? Or section 2.5 seems to suggest some other simulation was done where DustCOMM was used to adjust the emission fluxes, but I can't really tell which or if indeed there are actually seven independent simulations. Presuming this is a non-radiatively interactive CTM (which use of MERRA-2 meteorology suggests) then maybe there are two simulations, the "Default" and one performed with an adjust at emission particle size distribution? Or otherwise I don't understand: in any of experiments 2, 4-7 are you just using DustCOMM directly for the

calculations? How does the model get relaxed to the information from DustCOMM to simultaneously adjust the loading, 3D distribution, and particle size distribution if it does not actually just become the DustCOMM result?

**Response:**

The DustCOMM data set can be handled in offline calculations performed with different refractive indices and aerosol concentrations (Ito et al., 2018 and references therein). However, since the dust is highly episodic, we did not use the seasonally averaged DustCOMM directly for the calculations in the off-line model. The bias in the size-resolved dust emission was corrected in the CTM simulations, as was described on p.9, l.195 (Ito et al., in review, 2021). The bias in the 3-D particle size distribution was corrected prior to calculating the radiative fluxes using the RRTMG, as was described on p.9, l.205 (Ito et al., in review, 2021). We elucidated them on p.4, l.110, p.8, l.218, p.8, l.224, and p.8, l.228:

"Five sensitivity experiments were handled in the RRTMG calculations performed with different refractive indices and hourly averaged aerosol concentrations with the two chemical transport model simulations of "IMPACT" and "DustCOMM". The other three experiments were calculated from the model output with a post-processor."

"To correct the bias in the seasonally averaged size-resolved dust emission in the IMPACT model, the sum of bin 1, bin 2, and bin 3 dust emission flux was scaled by the seasonal mean of the ratio of the sum of bin 1, bin 2, and bin 3 dust column loading between the model and DustCOMM at each 2-D grid box."

"To adjust the size bias in dust emissions, the mass fraction of emitted dust for each bin was prescribed according to the size-resolved total column loading of DustCOMM at each 2-D grid box."

"To correct the bias in the seasonally averaged 3-D dust size distribution after the transport, the mass fraction of dust concentration for each bin between 0.2 and 20.0 µm of diameter was scaled at each 3-D grid box prior to calculating the radiative fluxes using the RRTMG by the ratio of mass concentration of $PM_{2.5}$ (i.e., the sum of bin 1 and bin 2) to each bin (Table 3)."

**Comment 3:**

A lot of the discussion of refractive indices seems disorganized, with something covered in 2.2 and some at the end of section 2.5. Could this be consolidated in a single subsection that goes over the refractive indices? You would also make a useful contribution if you included a figure in the paper that showed the spectral refractive index for your different choices. For the mineralogical map in Figure 1 and used in experiments 1 and 6 are you just applying different look up tables in different regions? You are not actually tracking mineral composition in the forward simulation, are you?

**Response:**

We added the section of "Spectral refractive index and sensitivity experiments to mineralogical compositions" and the plots of the spectral refractive index to Fig. 1, as was suggested. The global mean of the mineral composition was used in Fig. 1. To avoid the confusion (refractive index shown in Figure 1 was different from that used in Experiments 1 and 6), the regional mean (previous Experiment 5 used the refractive index shown in Figure 1) was removed in revised paper. The tagged tracer was used for each regional and each mineral source. This was described on p.5, l.144:

"To derive atmospheric concentration of mineral composition for dust aerosol, "tagged" tracer was used for each size-resolved mineral source."

**Comment 4:**

There are three different regional numbering schemes at use in the paper (Figure 1, Figure 3, and Table 4). Most challenging is differences in Figure 3 and Table 4, which they are similar in number and similar in sense though not geographic layout. The reader must hold several pieces of paper to follow all of this.

**Response:**

The regional number for clear-sky dust radiative effect efficiency in Table 4 was not shown in Figure 3, which showed $DAOD_{550}$. We labeled S# for source in Figure 1, A# for aerosol optical depth in Figure 3, and R# for radiative effect efficiency in Table 4, respectively. The figure captions were revised for Figure 1 and Figure 3.

**Comment 5:**

I don't follow the discussion of Figure 9. I *think* what is being shown is something in the sense of differences between simulations as in Table 2, but this isn't clearly the case to me. There are missing symbols in (a) and (b) which are maybe because some of the simulations are degenerate with others, hard to tell. It seems stated that the differences from the default are shown, but confusingly the default simulation is still shown and I don't know what the "Baseline" simulation is.

**Response:**

Figure 9 was separated into 2 figures of Figs. 9 and 10.

**Minor comments**

**Comment 1:**

Line 99: Is this a CTM (i.e., there is no radiative feedback of the dust on the simulation itself)? Please state that clearly here.

**Response:**

Yes, it is. The radiative feedback can be predicted by a separate version of the IMPACT coupled to the NCAR CAM5.3 model (Penner et al., 2018). We stated it on p.5, l.125:

"The chemical transport model was driven by the Modern Era Retrospective analysis for Research and Applications 2 (MERRA-2) reanalysis meteorological data from the National Aeronautics and Space Administration (NASA) Global Modeling and Assimilation Office (GMAO) (Gelaro et al., 2017). The radiative feedback of the dust on the climate model simulation can be predicted by a separate version of the model (Penner et al., 2018)."

**Comment 2:**

Line 104: Is this a bulk aerosol scheme, modal or something else? The dust appears to be treated in a sectional scheme.

**Response:**

There are different versions of the IMPACT model, as was mentioned in our response to minor comment 1. In this version of the IMPACT model, two modes are used for sulfate aerosol (nuclei and accumulation mode), and two moments are predicted within each mode (sulfate aerosol number and mass concentration) (Liu et al., 2015). The dust is represented according to the sectional scheme, as was mentioned. We added following sentences to clarify the version of the IMPACT model on p.5, l.133, p.6, l.179:

"In this version of the IMPACT model, two modes were used for sulfate aerosol (nuclei and accumulation mode), and two moments were predicted within each mode (sulfate aerosol number and mass concentration) (Liu et al., 2015)."

"The mineral dust particles were assumed to follow prescribed size distributions within each size bin (Liu et al., 2015). In applying the look-up table, the size spectrum for mineral dust was divided into 30 sub bins (Wang and Penner, 2009)."

**Comment 4:**

Line 110: If the default scaling factor for IMPACT dust emissions is determined from observations why is the AOD so low? Is this because that is a constraint on the source regions but not downstream?

**Response:**

It is a constraint on dust-dominated AERONET stations which are mostly located in North African regions (Kok et al., 2014), as was described on p.5, l.110. Thus, the model could underestimate the global and annual mean $DAOD_{550}$, due to seasonally and regionally underestimates. In revised paper, we showed the differences of DAOD between the model simulations in Fig. 3. As was shown in Fig. 3, Tables S1 and S2, DAOD from the IMPACT simulation was lower than the semi-observationally-based data over East Asia and Bodele/Sudan in winter. We added following sentences on p.11, l.320:

"The lower $DAOD_{550}$ from E2 than E1 was mostly found over East Asia and Bodele/Sudan in winter (Fig. 2, Table S2)."

**Comment 5:**

Line 126-128: Sentence beginning "Consequently..." does not make sense to me. What ensemble of model results are you referring to? Are you just saying that even adding this asymmetric correction to the IMPACT simulation it still doesn't give a good dust lifetime versus DustCOMM? Maybe you mean "Nevertheless" instead of "Consequently?"

**Response:**

Kok et al. (2017) constrain the globally averaged size-resolved dust lifetime from the lifetime simulated with nine different climate and chemical transport models. These include GISS, GMOD, CESM, MOZART, UMI, MERRAero, WRF-Chem, GEOS-Chem, and HadGEM. This is the one of our main topics and the reason why we adjusted the size-resolved abundance of dust concentration. The impact of this on the results and overall sensitivity was evaluated by "Experiment 3 – Experiment 4", as was summarized in Table 2. To elucidate this, we added the following sentence on p.6, l.164:

"Nevertheless, the lifetime of the dust aerosol for the largest-size bin in the IMPACT model, even after accounting for asphericity (1.4 days for 5–20 μm of diameter), was significantly shorter than an ensemble of model results (2.1 ± 0.3 days for the mass mean diameter of 8.3 μm) (Kok et al., 2017). The impact of this underestimate of atmospheric lifetime is explored using the DustCOMM data set, as was summarized in Table 2 (E3 – E4)."

**Comment 6:**

Lines 137 - 146: I think you are describing here the construction of the "Mineralogical map" refractive indices in the shortwave. The referencing to Table 1 simulation is again confusing: "default simulations (denoted as'mineral')" — no, default is not denoted as "mineral" it is denoted as "default." Similarly "Default" is also using "global" LW optics. I wonder if some other coding schemes for experiments would just make this a little less twisty to follow.

**Response:**

We rephrased the short name by a long name and used Experiment number such as E1. We revised the terms in Table 1.

**Comment 7:**

Line 148: You refer to internally mixing dust with other components in the radiative forcing calculations. How are those other species partitioned across the four dust size bins? How does this assumption project onto the calculations you are doing with the non-spherical optics? I would think that internal mixing drives particles toward sphericity, but clearly you are doing some calculations with asymmetric dust treatment. Do the non-dust portions also get this non-spherical treatment? None of this clear.

**Response:**

The surface coating of sulfate on dust aerosols occurs as a result of the condensation of sulfuric acid gas on their surfaces, coagulation with sulfate aerosol, and formation in aqueous reactions within cloudy regions of the atmosphere (Liu et al., 2015). The heterogeneous uptake of nitrate,

ammonium, and water vapor by each aerosol for each size bin is interactively simulated in the model following a hybrid dynamical approach (Feng and Penner, 2007). The chemical processing can form a uniform coating around the mineral core and therefore decrease particle asphericity during transport. This is implicitly considered in the simple shape distribution of dust, because Huang et al. (2019) averaged all medians and geometric standard deviations to obtain the globally averaged shape distributions of dust aerosols. We used the adjustment factors for the dust particles, as was described on p.8, l.184. To elucidate this, we added the following sentences on p.5, l.135 and p.8, l.235:

"The surface coating of sulfate on dust aerosols occurred as a result of the condensation of sulfuric acid gas on their surfaces, coagulation with sulfate aerosol, and formation in aqueous reactions within cloudy regions of the atmosphere (Liu et al., 2015). The heterogeneous uptake of nitrate, ammonium, and water vapor by each aerosol for each size bin was interactively simulated in the model following a hybrid dynamical approach (Feng and Penner, 2007).."

"The atmospheric aging of mineral dust can form a uniform coating around the mineral core and therefore decrease particle asphericity during transport. This is implicitly considered in the globally averaged shape distribution of dust (Huang et al., 2019)."

**Comment 8:**

Line 150: Adjustment factors are noted to account for the missing treatment of LW scattering. But it is not stated *what* is adjusted? The overall fluxes? Please clarify this.

**Response:**

We added the "LW radiative fluxes" and moved after the flux calculation on p.7, l.186:

"As the LW scattering was not accounted for in the RRTMG, we multiplied the LW radiative fluxes by the adjustment factors of $1.18 \pm 0.01$ and $2.04 \pm 0.18$ for the dry particles at the surface and TOA"

**Comment 9:**

Line 155: Again to the internal mixing: how is the dust radiative effect separated out from internally mixing particles? Is it the difference of (dust+rest) internal mix and dust only?

**Response:**

Five types of aerosols were assumed to be externally mixed in each size bin for the computation of spectral optical properties (Xu and Penner, 2012) on p.5, l.139:

"Five types of aerosols (i.e., dust, nucleated sulfate, carbonaceous aerosols from fossil fuel combustion, carbonaceous aerosols from biomass burning, and sea salt) were assumed to be externally mixed in each size bin for the computation of spectral optical properties (Xu and Penner, 2012)."

**Comment 10:**

Line 170: "to bias adjust an ensemble"

**Response:**

This was corrected on p.7, l.203:

"which is used subsequently to adjust the bias in an ensemble of six global model simulations"

**Comment 11:**

Line 283: Reference I think is to Figure S2f.

**Response:**

This was deleted, as was suggested by the reviewers.

**Comment 12:**

Figure 3: Caption references a non-existent panel (d).

**Response:**

This was deleted, as was pointed out.

**Comment 13:**

*(check not a printer issue...) Figure 3: I am having a hard time reconciling the coloration of the regions comparing (a) with (b) and (c) with the numerical values in Table S1. For example, region 4 (Mali/Niger) has a stated semi-observationally-based AOD of 0.462 in Table S1 which ought to be a red color, but the region is plotted yellow. This confuses the discussion about the "goodness" of the simulations. Similar I think for regions 2 and 9.

**Response:**

You are right that this was confusing because Figure 3 represented the annually mean for semi-observationally-based data and seasonally mean for model results. The caption and figure were corrected. In revised paper, we showed the evaluation of the various model experiments against semi-observationally-based estimates in box plots and Taylor diagrams (Taylor, 2001) to provide a concise statistical summary of the bias, correlation coefficient, root mean square errors, and the ratio of standard, as was described in our response to general comment 3 by reviewer #1.

**Comment 14:**

Figures 4 - 8, S3, S5: The units for the radiative effect efficiency (stated at bottom of each figure) should. properly be written (W m-2 AOD-1)

**Response:**

This was corrected in Figs. 4 and 5, as was pointed out. Figs. 6, 7, and 8 represented radiative effect (W·m$^{-2}$). Figs. S3 and S5 were deleted.

**Comment 15:**

Figure 4: The SW TOA side is a bit misleading as the color bar ends at zero on the right, but from the tables clearly the values go to positive numbers. Might consider expanding color bar.

**Response:**

This was expanded, as was pointed out.

**Comment 16:**

Figure 6 - 8: The word "Atmosphere" is misspelled in panel (c) in each.

**Response:**

This was corrected, as was pointed out.

**Comment 17:**

Figure 9: Where is the "+" symbol in (a)? Where is the orange star symbol (Improved simulation) in (b)?

**Response:**

These were overlapped. Figure 9 was separated into 2 figures of Figs. 9 and 10.

**Comment 18:**

Tables S1, S2: Please add a leading column for the region number so the reader can associate the location described with the map.

**Response:**

This was done, as was suggested.

**Comment 19:**

Table S3: third line, should "69" be "-69"?

**Response:**

This was corrected, as was pointed out. We note that the values for model estimates in previous Tables S3–S6 were all sky radiative effect efficiencies and corrected to clear-sky values.

---

## Referee Report (RR1)

This study uses the observation-based size-resolved dust concentration, asphericity factor and spectral refractive to improve the simulated dust radiative effect at TOA, surface and in the atmosphere. The adjustment improves the agreement of simulated globally and annually averaged DAOD with the semi-observational based estimate. Several experiments are implemented to investigate the sensitivity of dust radiative effect to dust size, shape, and refractive index. This study finds that a less absorptive SW dust refractive index (RI) and more absorptive LW RI are required for coarse aspherical dust to achieve a better agreement against a semi-observational-based radiative effect efficiency at TOA. The combination of coarse aspherical dust with less absorptive SW refractive index induces a less heating effect in the atmosphere. Overall, this study is interesting and comprehensive, it provides new insights to the research community. I have several minor comments listed below.

1. Line220: In this section, I think it would be helpful to add some equations to show how exactly the emission flux and emitted dust mass fraction are scaled.
2. Line267: Should 'DB17' be 'DB19'? Please correct
3. Line272: Should 'DB19' be 'DB17'? Please correct
4. Line320: In 'The **lower** DAOD550 from E2 to E1', lower should be higher, right?
5. Line376: Please change 'V83 simulation' to 'IMPACT-Sphere-Mineral-V83' to be specific, 'V83 simulation' is confusing, since both E1 and E2 use V83.
6. Section 3.5. Figure 9 shows the comparison between E2 and E1, the difference between E2 and E1 could be divided into 3 aspects: dust size, shape, and RI (SW-RI). Then each aspect is investigated in figure 10. However, I wonder why to choose E4 as a reference case in figure 10? We could see that LW-RI (volz83) is the same for E2 and E1, there is not an experiment (or marker) in figure 10 could directly illustrate the contribution of RI difference (only SW-RI is different in this case) to the radiative effects difference between E1 and E2. Would it be better if change the reference case to be E2? Then set up three experiments to change size, shape, SW-RI respectively?
7. The caption of Figure 4 and Figure 5 is not consistent with the figure, please correct.
8. I think the 'Sphericity' in Figure 10 is the 'Aspherical shape' experiment (E5-E4) in table2. It would be easier to understand if the names are consistent. In addition, in figure10, it would be clearer if add the experiment number (from Table 2) for each marker.

---

## Author Response (AR2)

Our responses to the comments from Reviewer 2 are included in the followings.

Reviewer #2

**Comments:**

The paper presents a model-based sensitivity analysis of various factors that affect the calculation of dust direct radiative forcing in both SW and LW parts of the spectrum. Some of the model simulations are partially constrained by prior results from a semi-observationally-based dust climatology, which in particular modifies the simulated fine- and coarse-mode apportioning of the dust, as well as the overall loading.

The paper is improved in its revision, and I remain convinced there is a kernel of something really useful here, but it is still confusing in its organization and layout with a disturbing number of typographical errors that need to be cleaned up. I cannot recommend the publication of this paper at this point. It also seems to have lost its main thrust or conclusion, with no clear recommendation of a path forward evident. It seems squarely stuck in model sensitivity analysis land.

**Response:**

We would like to thank the reviewer for his or her constructive comments. The reviewer understands the simulations correctly that we did not use climate model (coupled IMPACT/CAM) but used CTM (coupled IMPACT/RRTMG) as in the initial submission for the revised manuscript. Accordingly, we did not change the main conclusion from the initial submission but revised the readability of our manuscript following suggestions made by the reviewers and the editor.

**Main points**

**Comment 1:**

Methods and modeling

I still don't understand how many different CTM simulations were performed.

Line 106 says two IMPACT simulations were performed "with the finer dust size", while line 112 refers to two CTM simulations with "IMPACT" and "DustCOMM" configurations. I suspect it is the latter. Elsewhere it is variously referring to eight experiments and five sensitivity experiments. I suspect actually there are two independent simulations (E1 and E2) that have different a priori presumptions of the dust particle size distribution and emission fluxes and that result in a database of three dimensional, time varying dust distributions (I justify this also by seeing only E1 and E2 report masses in Table 3). Further, I infer that both simulations actually account for dust asphericity in calculation of settling velocities and that the distinction between spherical and non-spherical dust is only in the a posteriori radiative fluxes and AOD. Please clarify.

**Response:**

The reviewer understands the simulations correctly. We stated chemical transport (IMPACT) model and radiative transfer (RRTMG) module simulations more clearly on p.4, l.104.

"We examined the dust radiative effects using ten combinations of different numerical experiments that varied (1) the simulated dust concentration and their size distribution, (2) particle shape, and (3) mineralogical composition (Tables 1 and 2). Two RRTMG calculations used the hourly averaged aerosol concentrations calculated from one IMPACT model simulation (E1 and E3) (denoted as "IMPACT"). The two sensitivity experiments were handled in the RRTMG calculations performed with the distinction between spherical and non-spherical dust and different refractive indices. We denoted "Sphere" when the RRTMG calculations used the spherical assumption on the particle shape, while the IMPACT model considered asphericity in calculation of gravitational settling velocities. On the other hand, we denoted "Asphere" when the dust asphericity was also considered in the RRTMG calculations."

**Comment 2:**

These are then used in the offline/separate radiative transfer calculator (line 128). I suspect the remaining 8 sensitivity experiments are using the distributions from E1 and E2 and just applying the different optical property assumptions. This needs to be clarified in the paper. (Also, line 128 states: "the radiative feedback of the dust on the climate model simulations can be predicted…" what climate model simulation? I think you just mean you have an offline tool for doing the radiative effect calculation. These are CTM experiments and there is no feedback on the simulated dust distributions. If I've got that wrong then I am thoroughly confused about what you actually did in this study. I think you are only diagnosing radiative effects, not looking at feedbacks.

**Response:**

The reviewer is right that the NCAR Community Atmosphere Model (CAM) was not used in this study. We revised the sentence to avoid the confusion on p.4, l.111 and p.5, l.135.

"Subsequently, the simulated dust concentration and the size distribution were adjusted to the semi-observationally-based concentrations (Adebiyi and Kok, 2020) in another chemical transport model simulation, which was performed with the five RRTMG calculations (E4, E5, E6, E8, and E9) (denoted as "DustCOMM"). The term "semi-observationally-based" is used for DustCOMM, $DAOD_{550}$, and dust radiative effect efficiency when the estimates are based on the combination of observations and models. We examined different refractive indices for the dust mineralogy to represent the regional variations in refractive indices (denoted as "Mineral", "DB17", "DB19", "V83", "Less SW", "More LW", "More SW", and "Less LW"). Thus, the other three experiments (E2, E7, and E10) were calculated from the model output with a post-processor. DustCOMM-Asphere-DB19-V83 (E2) was obtained from combination of DustCOMM-Asphere-DB19-DB17 (E4) for SW and DustCOMM-Asphere-Mineral-V83 (E6) for LW. DustCOMM-Asphere-Less-More (E7) was obtained from combination of DustCOMM-Asphere-Less-Less (E8) for SW and DustCOMM-Asphere-More-More (E9) for LW. DustCOMM-Asphere-More-Less (E10) was obtained from combination of DustCOMM-Asphere-More-More (E9) for SW and DustCOMM-Asphere-Less-Less (E8) for LW."

"Thus, the radiative feedback of the dust aerosol on the climate was not considered in this study."

**Comment 3:**

For clarity I suggest you move line 142 and following text "Dust emissions were dynamically simulated…" to following line 132 "….their precursor gases." This would clarify the model description.

**Response:**

This is done.

**Comment 4:**

Consistent usage of reference to Liu et al. 2015 throughout paper I think s/b Liu et al. 2005. Please correct or else provide citation.

**Response:**

This is corrected.

**Comment 5:**

There is further some inconsistency in description of optical properties. Five aerosol species simulated are externally mixed for computation of spectral optical properties (line 141) but are later said to be internally mixed (line 182). I'm also confused about pulling out the dust-only signal on the computed fluxes. If the particles are internally mixed I would expect the radiative effect of a coated particle to not simply be the additive effect of uncoated dust + residual other particle contribution (line 185). Rather, would it make more sense to differentiate the dust-only properties (as the core) from the total coated particle? I think this is different than what is stated. In any case, if the point of the paper is to evaluate the dust radiative effect only (which it seems to be) then why make this distinction at all since everything appears to be calculated offline. I'm frustrated I still don't seem to understand what is being done in this paper.

**Response:**

This is mainly because the SW scattering effect of coating materials on aerosol cores is considered. The aerosol core species are externally mixed with each other, whereas each core is internally mixed with coating materials on each aerosol. Thus, the coated dust is externally mixed with the other coated aerosol core species in the RRTMG calculations. We did not change this treatment from previous studies (Xu and Penner, 2012), but revised the description only. The sentence is revised on p.7, l.188 and p.7, l.191.

"These coating materials on aerosol cores were treated as internally mixed with each aerosol core in each size bin. Thus, the coating materials on dust only can reduce solar absorption of mineral dust."

"The dust RE was estimated as the difference in the calculated radiative fluxes with all aerosols and with all aerosols except the dust coated with sulfate, nitrate, ammonium, and water for each bin."

**Comment 6:**

Line 247: should refer to "Aspherical" for consistency with Table 1.

**Response:**

Table 1 is corrected for consistency.

**Comments on Results section**

**Comment 7:**

Figure 1: the (b) and (d) figures render in pretty poor quality. There is no explanation in the caption for the presence of (b) and (d) and generally none of the parts are referred to adequately in the caption.

**Response:**

Fig.1 and its caption are revised.

"Imaginary part of the refractive index at (a) 0.52 μm, (b) SW, (c) 9.7 μm, and (d) LW."

**Comment 8:**

Figure 2: the lengthy caption could be greatly reduced as the regional boundaries are all present in Table S1. Also the caption and Table S1 refer to the regions as "A1" etc. but in the (a) panel they are labeled as "S1" etc.

**Response:**

Fig.2 and its caption are revised.

"The coordinates and the values of $DAOD_{550}$ at the 15 regions (marked in Fig. 2a) in summer were listed in Table S1."

**Comment 9:**

Figure 4: the caption does not describe the figure as presented. There are panels (a) through (j) and the captioning breaks down by the time you get to the description of panel (g). Also could label shown regions as in Figure 2 with reference to the regions in tables S2-S6

**Response:**

This is revised.

**Comment 10:**

Figure 5: same comment as for Figure 4

**Response:**

This is revised.

**Comment 11:**

Figure 10: the clarity of this figure would be enhanced if in addition to the legend of symbols you wrote explicitly the experiments being differenced as in Table 2. I.e., the black hexagon could state next to it (E3-E4)

**Response:**

This is revised.

**Comment 12:**

Line 320: this is unclear or incorrect. As presented in Figure 2 and Table 3 I expect that E2 has > dust AOD than E1. What's written seems to say opposite.

**Response:**

This sentence and Fig. 2(c) are revised and combined with the sentence to represent the higher $DAOD_{550}$ from E2 than E1 (E2 – E1) on p.12, l.341.

"We also found higher $DAOD_{550}$ from E2 than E1 over East Asia and Bodele/Sudan in winter (Fig. 2, Table S2)."

**Comment 13:**

Line 335: The statement about comparisons for other regions does not appear to refer to Figure 4 actually. No idea what is meant here.

**Response:**

This is deleted.

**Comment 14:**

Line 385: Why is E4 going forward the baseline simulation to compare against? Not sure it's a wrong choice, but it just seems out of context when E2 seems the preferred simulation.

**Response:**

We thank both the reviewers for this insightful comment. We choose E4 as a reference case in figure 10 to show the model sensitivity to LW RI used in other studies in Fig. 9, as was stated on p.11, l.311 and suggested by the reviewer #1 (https://doi.org/10.5194/acp-2021-134-RC1 and https://doi.org/10.5194/acp-2021-134-AC1). Specifically, both Di Biagio et al. (2020) and Balkanski et al. (2021) used the 'DB17' and considered dust with diameters in excess of 20 μm. We revised the sentences to clarify it on p.14, l.393, p.14, l.401, and p.14, l.429:

"To elucidate the differences in dust radiative effects between the IMPACT-Sphere-Mineral-V83 (E1) and DustCOMM-Asphere-DB19-V83 (E2) simulations and to explore the variability in different previous model estimates (Fig. 9), the differences in annually averaged radiative effects of mineral dust from DustCOMM-Asphere-DB19-DB17 (E4) simulation were shown in Fig. 10."

"This revision can be divided into (1) the size-resolved abundance (black hexagons, E3 – E4, in Fig. 10), (2) SW refractive index (red diamonds, E6 – E4, in Fig. 10), and (3) particle shape (red circles, E5– E4, in Fig. 10). Additionally, we show the sensitivity of dust RE to LW refractive index (DB17), which was used by both Di Biagio et al. (2020) and Balkanski et al. (2021)."

"At the same time, both Di Biagio et al. (2020) and Balkanski et al. (2021) used DB17 and considered dust with diameters more than 20 μm. Thus, the more absorptive LW dust refractive index (V83, E6 for LW: 1.00 W·m$^{-2}$) than DB17 (E4 for LW: 0.58 W·m$^{-2}$) (E6 – E4 for LW: 0.42 W·m$^{-2}$ at the horizontal axis of red diamond in Fig. 10b) could also contribute to the less surface cooling, which might be partially compensated for in our model by the omission of dust with diameters more than 20 μm."

**Comment 15:**

Line 393: Statement here is also unclear. I think E6 is mineralogical map referred to here which I think is more absorbing than E4, not less.

**Response:**

The reviewer is right that E6 is more absorbing than E4, but the positive (negative) value means less (more) cooling. To avoid the confusion, the sentence is revised on p.14, l.407.

"Second, this less SW cooling effect with coarser dust (E3 – E4) was partially compensated for by more SW cooling with the use of the less absorptive SW refractive index (E4: –0.32 W·m$^{-2}$) than E6 (0.02 W·m$^{-2}$)."

**Comment 16:**

Line 442: Where are these numbers from? They don't appear in Table 5.

**Response:**

The numbers obtained from Balkanski et al. (in review) were updated to the final publication (Balkanski et al., 2021) in the revised paper.

**Comment 17:**

Line 447: Abstract refers to -0.60 W m-2 where here you say -0.88 W m-2. Abstract agrees with Table 5.

**Response:**

This is corrected.

**References**

Balkanski, Y., Bonnet, R., Boucher, O., Checa-Garcia, R., and Servonnat, J.: Dust induced atmospheric absorption improves tropical precipitations in climate models, Atmos. Chem. Phys. Discuss. [preprint], https://doi.org/10.5194/acp-2021-12, in review, 2021.

Balkanski, Y., Bonnet, R., Boucher, O., Checa-Garcia, R., and Servonnat, J.: Better representation of dust can improve climate models with too weak an African monsoon, Atmos. Chem. Phys., 21, 11423–11435, https://doi.org/10.5194/acp-21-11423-2021, 2021.

Di Biagio, C., Balkanski, Y., Albani, S., Boucher, O., and Formenti, P.: Direct radiative effect by mineral dust aerosols constrained by new microphysical and spectral optical data. Geophys. Res. Lett., 47, e2019GL086186, https://doi.org/10.1029/2019GL086186, 2020.

Xu, L. and Penner, J. E.: Global simulations of nitrate and ammonium aerosols and their radiative effects, Atmos. Chem. Phys., 12, 9479–9504, https://doi.org/10.5194/acp-12-9479-2012, 2012.

Our responses to the comments from Reviewer 3 are included in the followings.

Reviewer #3

**Comments:**

This study uses the observation-based size-resolved dust concentration, asphericity factor and spectral refractive to improve the simulated dust radiative effect at TOA, surface and in the atmosphere. The adjustment improves the agreement of simulated globally and annually averaged DAOD with the semi-observational based estimate. Several experiments are implemented to investigate the sensitivity of dust radiative effect to dust size, shape, and refractive index. This study finds that a less absorptive SW dust refractive index (RI) and more absorptive LW RI are required for coarse aspherical dust to achieve a better agreement against a semi-observational-based radiative effect efficiency at TOA. The combination of coarse aspherical dust with less absorptive SW refractive index induces a less heating effect in the atmosphere. Overall, this study is interesting and comprehensive, it provides new insights to the research community. I have several minor comments listed below.

**Response:**

We would like to thank the reviewer for his or her constructive comments which helped us to improve the readability of our manuscript substantially. We revised the manuscript following suggestions made by the reviewer.

**Comment 1:**

1. Line220: In this section, I think it would be helpful to add some equations to show how exactly the emission flux and emitted dust mass fraction are scaled.

**Response:**

Three equations are added for the bias correction factor (Eq. 1), mass fraction for each size bin (Eq. 2), and dust emission flux after the adjustment (Eq. 3).

**Comment 2:**

2. Line267: Should 'DB17' be 'DB19'? Please correct

**Response:**

This is corrected.

**Comment 3:**

3. Line272: Should 'DB19' be 'DB17'? Please correct

**Response:**

This is corrected.

**Comment 4:**

4. Line320: In 'The lower DAOD550 from E2 to E1', lower should be higher, right?

**Response:**

This sentence and Fig. 2(c) are revised and combined with the sentence to represent the higher DAOD$_{550}$ from E2 than E1 (E2 – E1) on p.12, l.341.

"We also found higher DAOD$_{550}$ from E2 than E1 over East Asia and Bodele/Sudan in winter (Fig. 2, Table S2)."

**Comment 5:**

5. Line376: Please change 'V83 simulation' to 'IMPACT-Sphere-Mineral-V83' to be specific, 'V83 simulation' is confusing, since both E1 and E2 use V83.

**Response:**

This is changed.

**Comment 6:**

6. Section 3.5. Figure 9 shows the comparison between E2 and E1, the difference between E2 and E1 could be divided into 3 aspects: dust size, shape, and RI (SW-RI). Then each aspect is investigated in figure 10. However, I wonder why to choose E4 as a reference case in figure 10? We could see that LW-RI (volz83) is the same for E2 and E1, there is not an experiment (or marker) in figure 10 could directly illustrate the contribution of RI difference (only SW-RI is different in this case) to the radiative effects difference between E1 and E2. Would it be better if change the reference case to be E2? Then set up three experiments to change size, shape, SW-RI respectively?

**Response:**

We thank both the reviewers for this insightful comment. We choose E4 as a reference case in figure 10 to show the model sensitivity to LW RI used in other studies in Fig. 9, as was stated on p.11, l.311 and suggested by the reviewer #1 (https://doi.org/10.5194/acp-2021-134-RC1 and https://doi.org/10.5194/acp-2021-134-AC1). Specifically, both Di Biagio et al. (2020) and Balkanski et al. (2021) used the 'DB17' and considered dust with diameters in excess of 20 μm. We revised the sentences to clarify it on p.14, l.393, p.14, l.401, and p.14, l.429:

"To elucidate the differences in dust radiative effects between the IMPACT-Sphere-Mineral-V83 (E1) and DustCOMM-Asphere-DB19-V83 (E2) simulations and to explore the variability in different previous model estimates (Fig. 9), the differences in annually averaged radiative effects of mineral dust from DustCOMM-Asphere-DB19-DB17 (E4) simulation were shown in Fig. 10."

"This revision can be divided into (1) the size-resolved abundance (black hexagons, E3 – E4, in Fig. 10), (2) SW refractive index (red diamonds, E6 – E4, in Fig. 10), and (3) particle shape (red circles, E5– E4, in Fig. 10). Additionally, we show the sensitivity of dust RE to LW refractive index (DB17), which was used by both Di Biagio et al. (2020) and Balkanski et al. (2021)."

"At the same time, both Di Biagio et al. (2020) and Balkanski et al. (2021) used DB17 and considered dust with diameters more than 20 μm. Thus, the omission of dust with diameters more than 20 μm could also contribute to the less surface warming in our model, which might be partially compensated for with the use of the more absorptive LW dust refractive index V83 (E6: 1.00 $W \cdot m^{-2}$) than DB17 (E4: 0.58 $W \cdot m^{-2}$) (E6 – E4: 0.42 $W \cdot m^{-2}$ at the horizontal axis of red diamond in Fig. 10b)."

**Comment 7:**

7. The caption of Figure 4 and Figure 5 is not consistent with the figure, please correct.

**Response:**

This is corrected.

**Comment 8:**

8. I think the 'Sphericity' in Figure 10 is the 'Aspherical shape' experiment (E5-E4) in table2. It would be easier to understand if the names are consistent. In addition, in figure10, it would be clearer if add the experiment number (from Table 2) for each marker.

**Response:**

This is done.

**References**

Balkanski, Y., Bonnet, R., Boucher, O., Checa-Garcia, R., and Servonnat, J.: Better representation of dust can improve climate models with too weak an African monsoon, Atmos. Chem. Phys., 21, 11423–11435, https://doi.org/10.5194/acp-21-11423-2021, 2021.

Di Biagio, C., Balkanski, Y., Albani, S., Boucher, O., and Formenti, P.: Direct radiative effect by mineral dust aerosols constrained by new microphysical and spectral optical data. Geophys. Res. Lett., 47, e2019GL086186, https://doi.org/10.1029/2019GL086186, 2020.

Our response to the comment from Reviewer 4 is included in the following.

Reviewer #4

**Comments:**

The authors did a great work in revising the manuscript. I have no further comments.

**Response:**

We would like to thank the reviewer for his or her constructive comments.

---

## Author Response (AR3)

Dear Dr. Burrows:

Thank you for handling the paper nicely. I uploaded the manuscript with technical corrections in figures.

Best regards,
Akinori Ito
JAMSTEC